# Spatial distributions of $X_{CO_2}$ seasonal cycle amplitude and phase over northern high latitude regions

Nicole Jacobs[1], William R. Simpson[1], Kelly A. Graham[2], Christopher Holmes[2], Frank Hase[3], Thomas Blumenstock[3], Qiansi Tu[3], Matthias Frey[3,4], Manvendra K. Dubey[5], Harrison A. Parker[5,6], Debra Wunch[7], Rigel Kivi[8], Pauli Heikkinen[8], Justus Notholt[9], Christof Petri[9], and Thorsten Warneke[9]

[1]Department of Chemistry and the Geophysical Institute, University of Alaska Fairbanks, Fairbanks, AK, USA
[2]Department of Earth, Ocean, and Atmospheric Science, Florida State University, Tallahassee, FL, USA
[3]Karlsruhe Institute of Technology (KIT), Institute of Meteorology and Climate Research, Karlsruhe, Germany
[4]National Institute for Environmental Studies, Tsukuba, Japan
[5]Earth and Environmental Sciences, Los Alamos National Laboratory, Los Alamos, NM, USA
[6]California Institute of Technology, Pasadena, CA, USA
[7]Department of Physics, University of Toronto, Toronto, Canada
[8]Finnish Meteorological Institute, Sodankylä, Finland
[9]Institute of Environmental Physics, University of Bremen, Germany

**Correspondence:** William Simpson (wrsimpson@alaska.edu)

**Abstract.** Satellite-based observations of atmospheric carbon dioxide ($CO_2$) provide measurements in remote regions, such as the biologically sensitive but under sampled northern high latitudes, and are progressing toward true global data coverage. Recent improvements in satellite retrievals of total column-averaged dry air mole fractions of $CO_2$ ($X_{CO_2}$) from the NASA Orbiting Carbon Observatory 2 (OCO-2) have allowed for unprecedented data coverage of northern high latitude regions, while maintaining acceptable accuracy and consistency relative to ground-based observations, and finally providing sufficient data in spring and autumn for analysis of satellite-observed $X_{CO_2}$ seasonal cycles across a majority of terrestrial northern high latitude regions. Here, we present an analysis of $X_{CO_2}$ seasonal cycles calculated from OCO-2 data for temperate, boreal, and tundra regions, subdivided into 5° latitude by 20° longitude zones. We quantify the seasonal cycle amplitudes (SCA) and the annual half drawdown day (HDD). OCO-2 SCA is in good agreement with ground-based observations at five high latitude sites and OCO-2 SCA show very close agreement with SCA calculated for model estimates of $X_{CO_2}$ from the Copernicus Atmospheric Monitoring Services (CAMS) global inversion-optimized greenhouse gas flux model v19r1 and the CarbonTracker2019 model (CT2019B). Model estimates of $X_{CO_2}$ from the GEOS-Chem $CO_2$ simulation version 12.7.2 with underlying biospheric fluxes from CarbonTracker2019 (GC-CT2019) yield SCA of larger magnitude and spread over a larger range than those from CAMS, CT2019B, or OCO-2; however, GC-CT2019 SCA still exhibit a very similar spatial distribution across northern high latitude regions to that from CAMS, CT2019B, and OCO-2. Zones in the Asian Boreal Forest were found to have exceptionally large SCA and early HDD, and both OCO-2 data and model estimates yield a distinct longitudinal gradient of increasing SCA from west to east across the Eurasian continent. In northern high latitude regions, spanning latitudes from 47°N to 72°N, longitudinal gradients in both SCA and HDD are at least as pronounced as latitudinal gradients, suggesting a role for global atmospheric transport patterns in defining spatial distributions of $X_{CO_2}$ seasonality across these regions. GEOS-Chem surface

contact tracers show that the largest $X_{CO_2}$ SCA occurs in areas with the greatest contact with land surfaces, integrated over 15-30 days. The correlation of $X_{CO_2}$ SCA with these land surface contact tracers are stronger than the correlation of $X_{CO_2}$ SCA with the SCA of $CO_2$ fluxes or the total annual $CO_2$ flux within each $5°$ latitude by $20°$ longitude zone. This indicates that accumulation of terrestrial $CO_2$ flux during atmospheric transport is a major driver of regional variations in $X_{CO_2}$ SCA.

*Copyright statement.* TEXT

## 1  Introduction

The changing climate influences carbon exchange in every ecosystem on the planet and polar amplification is driving more rapid changes at higher latitudes (Smith et al., 2019; Park et al., 2018; Pithan and Mauritsen, 2014; Holland and Bitz, 2003; Manabe and Wetherald, 1975). An understanding of the rapidly changing carbon dynamics at high northern latitudes is nec-

essary to improve our understanding of global carbon exchange. However, despite the apparent importance of northern high latitude regions in quantifying the global carbon budget, Bradshaw and Warkentin (2015) point out that a great deal of uncertainty remains in the spatial patterns of carbon stocks and fluxes in Boreal Forest regions, and their results from predictive climate models show that the Boreal Forest may eventually shift from a carbon sink to a carbon source. Euskirchen et al. (2017), Barlow et al. (2015), and Pan et al. (2011) all point out that a shortage of observations in Boreal Forest regions is a

major impediment to understanding global carbon uptake, motivating further exploration of alternative data sources, such as satellite measurements. Since pioneering work by Thoning et al. (1989), analysis of the seasonal cycles of atmospheric $CO_2$ concentrations has been widely used to evaluate carbon exchange dynamics, and the amplitude of the regular seasonal oscillations in atmospheric $CO_2$ concentrations is a common metric used to infer relative $CO_2$ uptake. Many studies have combined process-based and atmospheric transport modeling with in situ and airborne observations to infer long-term temporal trends

and spatial distributions of seasonal $CO_2$ exchange, and concluded that Boreal Forest regions play an essential role in global carbon dynamics (Lin et al., 2020; Yin et al., 2018; Piao et al., 2017; Barlow et al., 2015; Bradshaw and Warkentin, 2015; Gauthier et al., 2015; Graven et al., 2013; Pan et al., 2011; Tans et al., 1990). Lin et al. (2020) compared seasonal cycle amplitudes (SCA) from surface in situ measurements of $CO_2$ to those estimated from GEOS-Chem transport modeling coupled with CAMS v17r1 flux estimates, and found that Siberia had the largest SCA of any region considered when normalized for

area. Furthermore, Lin et al. (2020) found that even though Siberia is a relatively small source region, fluxes from Siberia were the second most influential in determining SCA of in situ $CO_2$ on a global scale, following those from Northern Hemisphere midlatitudes.

It has been well established that the SCA of atmospheric $CO_2$ increases with latitude in the Northern Hemisphere due to the increased seasonal attenuation of sunlight which drives more extreme seasonality in temperature and ecosystem productivity

at higher latitudes. There is general consensus that this latitudinal gradient in SCA is increasing over time, so that while $CO_2$ SCA are increasing across the Northern Hemisphere, the SCA at higher northern latitudes are increasing at an accelerated rate.

There is still some controversy regarding what mechanisms are driving changes in $CO_2$ SCA and how spatial distributions or temporal trends in $CO_2$ seasonality are influenced by atmospheric transport patterns or regional changes in carbon exchange. Recent work by Liu et al. (2020) suggests that global increases in $CO_2$ SCA since the 1960's are a result of increases in growing season mean temperatures, and polar amplified warming would then explain the increase in the latitudinal gradients in SCA.

Studies by Piao et al. (2017), Forkel et al. (2016), and Graven et al. (2013) used global models to show that increasing latitudinal gradients in SCA are driven by the ecological effects of climate change and changes in vegetation, primarily suggesting $CO_2$ fertilization as the dominant mechanism. This point is confirmed by findings from Bastos et al. (2019) that attribute enhanced SCA in Boreal Asia and Europe to increases in net biome productivity as a result of $CO_2$ fertilization. Although they do not address the increase in latitudinal gradients over time, Zeng et al. (2014) and Gray et al. (2014) argue that agricultural expansion in the Northern Hemisphere midlatitudes has resulted in increases in seasonal carbon exchange, which, in turn, result in larger SCA of $CO_2$ concentrations on a global scale. Barnes et al. (2016) suggest that it is actually the Temperate Forest between $30°N$ and $50°N$ that is the dominant driver of seasonal carbon exchange on global scales. Yet another study by Yin et al. (2018) found evidence that challenged previous assumptions about the relationship between seasonal cycle amplitude and spring and autumn temperatures in northern high latitudes, emphasizing the need for continued data-driven model validation for these regions. Despite their disagreements, most agree that the seasonality in atmospheric $CO_2$ at northern high latitudes, and specifically the Boreal Forest, require continued attention as carbon dynamics continue to change. While this paper does not consider temporal changes in SCA, an assessment of spatial distributions of SCA implied by satellite-based observations over northern high latitude terrestrial regions can provide a good foundation for exploring temporal changes in these spatial distributions in later analyses.

Satellite-based infrared spectrometers like the NASA Orbiting Carbon Observatory 2 (OCO-2) (O'Dell et al., 2018; Wunch et al., 2017; Crisp et al., 2017), SCIAMACHY (Reuter et al., 2011; Bovensmann et al., 1999; Burrows et al., 1995), and GOSAT (Basu et al., 2013; Yoshida et al., 2013; Hamazaki et al., 2005) provide global measurements of column-averaged dry air mole fractions of $CO_2$ ($X_{CO_2}$), and particularly can quantify $X_{CO_2}$ in remote, un-instrumented regions. Retrievals and instrument technologies have been advancing rapidly, and boreal-forest-specific methods of $X_{CO_2}$ bias correction and quality control filtering have been developed and validated where ground truth exists (Jacobs et al., 2020b; Kiel et al., 2019; O'Dell et al., 2018). In addition, the development of collaborative networks of ground-based solar-viewing spectrometers, including the Total Carbon Column Observing Network (TCCON) and the Collaborative Carbon Column Observing Network (COCCON), has provided a framework for robust global validation of similar passive satellite-based observations (Frey et al., 2019; Wunch et al., 2011). These combined efforts of satellite-based and ground-based total atmospheric column measurements of $CO_2$ offer a wealth of opportunities for gaining insights into the global climate system as a whole.

In this manuscript, we quantify and analyze seasonal cycle parameters derived directly from satellite-based observations of $X_{CO_2}$, across the northern high latitude terrestrial regions. This work represents progress in the application of global monitoring of atmospheric $CO_2$ to the continued evaluation of global scale carbon dynamics, and shows how satellites like OCO-2 can be used to monitor $CO_2$ biospheric exchange. In this analysis, OCO-2 data over terrestrial northern high latitudes is used to explore spatial distributions of seasonal cycle amplitude (SCA) and seasonal cycle phase. Interpretation of these spatial dis-

tributions can be used to test previous claims and provide new insights into what is driving carbon exchange at northern high latitudes. In particular, we explore how seasonality in $X_{CO_2}$ differs for the North American, European, and Asian Boreal Forest regions, and how the Boreal Forest fits within the broader context of northern high latitude regions. In addition, seasonal cycle parameters derived from OCO-2 observations are combined with those from ground-based TCCON and COCCON observations, then compared with seasonal cycle parameters from three model frameworks: the Copernicus Atmospheric Monitoring Services (CAMS) global inversion-optimized greenhouse gas flux model estimates of $X_{CO_2}$ (Chevallier, 2020b), with in situ data assimilation, CarbonTracker2019 posterior $X_{CO_2}$ estimates (CT2019B; Jacobson et al. (2020a)), and the GEOS-Chem $CO_2$ simulation (Nassar et al., 2010) with underlying biospheric fluxes from CarbonTracker2019 (GC-CT2019). Ultimately, we use simulations of GEOS-Chem surface contact tracers and flux estimates from CAMS, CarbonTracker2019, as well as estimates of fossil fuel emissions from the Community Emissions Data System (CEDS; Hoesly et al., 2018) and estimates of biomass burning emissions from the Global Fire Emissions Database, Version 4.1 (GFED4.1s; Randerson et al., 2018; van der Werf et al., 2017) to address the question of how much spatial variability in $X_{CO_2}$ seasonal cycle parameters may be attributed to magnitudes of fluxes within the observation zones and how much may be attributed to the regional and continental scale accumulation of $CO_2$ fluxes during atmospheric transport.

## 2  Methods

### 2.1  OCO-2 data

The NASA Orbiting Carbon Observatory 2 (OCO-2) launched in 2014 and began collecting data in September of that year. Daily averages of $X_{CO_2}$ are calculated for each zone using observations from OCO-2 B9 Lite files (OCO-2 Science Team/Michael Gunson, Annmarie Eldering, 2018). Ongoing improvements in the ACOS retrieval algorithm and previous efforts by Jacobs et al. (2020b) to develop quality control thresholds tailored to OCO-2 B9 retrievals over Boreal Forest regions (Boreal QC) have allowed sufficient data over our $5°$ latitude by $20°$ longitude zones to construct $X_{CO_2}$ time-series that yield robust seasonal cycle parameterization. The Boreal QC was evaluated for use with terrestrial OCO-2 B9 retrievals north of $50°$N (Jacobs et al., 2020b), and the zones considered here cover the majority of land north of $50°$N. The southern boundaries of the southern-most zones of North America are at $47°$N, but the $3°$ of latitude is not expected to significantly impact the effectiveness of the Boreal QC filtering. Instead of the standard B9 bias correction, we use a modified bias correction that includes temperature at 700 hPa (T700), as discussed by Jacobs et al. (2020b), because it was found in previous results to reduce the seasonality of OCO-2 bias relative to ground-based TCCON and EM27/SUN measurements. Seasonal cycle fits to OCO-2 retrievals of $X_{CO_2}$ with the standard B9 bias correction, as well as fits to OCO-2 B10 retrievals, were also calculated and compared to model-derived seasonal cycle fits in the supplement (see Sect. S2). The spatial distribution of seasonal cycle parameters across northern high latitude regions is similar for all three types of OCO-2 retrievals, but the alternative bias correction yields improved agreement with model-derived seasonal cycle parameters.

## 2.2 TCCON and EM27/SUN data

The Total Carbon Column Observing Network (TCCON) is a ground-based network of sites observing $X_{CO_2}$ using high spectral resolution solar-viewing Fourier transform infrared spectrometers (FTS). Data are included from four TCCON sites: East Trout Lake, Canada in North American Boreal zone 3 (Wunch et al., 2018); Sodankylä, Finland in European Boreal zone 6 (Kivi et al., 2014; Kivi and Heikkinen, 2016); Białystok, Poland in European Temperate zone 2 (Deutscher et al., 2019); Bremen, Germany in European Temperate zone 3 (Notholt et al., 2019) (see site details in Table 1 and locations mapped in Fig. 1). The Collaborative Carbon Column Observing Network (COCCON) is a network of sites observing with the Bruker EM27/SUN FTS (Gisi et al., 2012), which are lower resolution mobile solar-viewing spectrometers that serve as complement to TCCON measurements. EM27/SUN observations have been compared to TCCON observations in multiple studies, most notably Sha et al. (2020), Tu et al. (2020), Frey et al. (2019), Velazco et al. (2018), and Hedelius et al. (2017). In most of these comparisons EM27/SUN and TCCON observations agree with biases less than 0.25 ppm on average. In some cases offsets between EM27/SUN and TCCON observations are reported to be as large as 2 ppm, but the proven stability of the EM27/SUN should allow for a bias correction that would yield good agreement between TCCON and EM27/SUN retrievals. The EM27/SUN instruments have measured $X_{CO_2}$ in a number of campaigns to validate OCO-2 and other satellite-based observations, including work by Jacobs et al. (2020b), Velazco et al. (2018), and Klappenbach et al. (2015), suggesting good agreement between EM27/SUN observations and satellite-based observations. In this analysis, observations with an EM27/SUN FTS in Fairbanks, Alaska, USA (65.859°N, 147.850°W; Jacobs et al. (2020a)) are used as a fifth ground-based comparison in the Boreal Forest. Fairbanks is an established COCCON site as of 2018, so the instrument participates regularly in performance and calibration checks at the central facility operated by KIT and data processed in compliance with COCCON recommendations are available. In this study, we use the GGG2014 retrieval algorithm coupled with the EM27/SUN GGG interferogram processing suite (EGI; Hedelius and Wennberg (2017)) instead of the standard COCCON retrieval methods for consistency with TCCON retrievals and because this data product has already been bias corrected to TCCON using side-by-side observations at Caltech, as described in detail by Jacobs et al. (2020b). Seasonal cycle fits for ground-based observations at these five sites use daily averages of retrievals collected within two hours of local solar noon, weighted by retrieval error, as described by Jacobs et al. (2020b). We refer to these daily averages as near noon ground-based (NNG) observations, and this time frame is chosen because OCO-2 overpasses tend to occur within approximately one hour of local solar noon over these northern high latitude regions.

## 2.3 Regions and zones

We define regions in North America, Europe, and Asia, which are further subdivided into 5° latitude by 20° longitude zones. For the purposes of this analysis, the classification of zones as temperate, boreal, or tundra, as well as the longitudinal division between the European and Asian regions are guided by maps of ecoregions from Hayes et al. (2011) and Euskirchen et al. (2007) (see Fig. 1). The Boreal Forest zones considered cover a narrower range of latitudes than the boreal regions defined for the Transcom 3 ecoregions (Gurney et al., 2000), which include all high Arctic tundra and portions of temperate Asia as part of the boreal regions. Otherwise, the North American Boreal region and Eurasian Boreal region defined by Transcom 3 are very

**Table 1.** Summary of instrument type, years of observations, geographic coordinates, and corresponding coordinates of the nearest model grid point in CAMS, CT2019B (CT), and GC-CT2019 (GC) for each ground site.

| Site | Type | Years | Site location | CAMS location | CT location | GC location |
|---|---|---|---|---|---|---|
| Bialystok | TCCON | 2014-2018 | 53.23°N, 23.03°E | 54.0°N, 22.5°E | 53.0°N, 22.5°E | 54.0°N, 22.5°E |
| Bremen | TCCON | 2014-2019 | 53.1°N, 8.85°E | 54.0°N, 7.5°E | 53.0°N, 7.5°E | 54.0°N, 10.0°E |
| East Trout Lake | TCCON | 2016-2019 | 54.35°N, 104.99°W | 54.0°N, 105.0°W | 55.0°N, 103.5°W | 54.0°N, 105.0°W |
| Sodankylä | TCCON | 2014-2019 | 67.26°N, 26.25°E | 67.37°N, 26.63°E | 67.0°N, 25.5°E | 68.0°N, 27.5°E |
| Fairbanks | EM27/SUN | 2016-2019 | 64.86°N, 147.85°W | 65.37°N, 146.25°W | 65.0°N, 148.5°W | 64.0°N, 147.5°W |

similar to the North American Boreal and Asian Boreal regions defined in this analysis. We differ markedly from Transcom 3 in defining separate European Boreal and European Temperate regions, while Transcom 3 combines all of Europe into a single European region. In North America, the zones are shifted by 3° latitude relative to the zones in Eurasia; starting at 47°N and extending in to 72°N in 5° increments. This was done to bring ground sites in the North American Boreal region closer to the center latitude of their encompassing zones and to more accurately fit the boundaries of temperate, boreal, and tundra biomes.

Also shown in Fig. 1 are the locations of five ground sites where long-term observations of $X_{CO_2}$ have been collected (see Table 1 for details). These ground sites include two sites in the European Temperate region (Białystok and Bremen), one site in the European Boreal region (Sodankylä), and two sites in the North American Boreal region (East Trout Lake and Fairbanks). Ground-based data are explained further in Sect. 2.2 and seasonal cycles of ground-based data are compared to satellite and model-derived seasonal cycles in Sect. 3.1.

## 2.4 $X_{CO_2}$ seasonal cycle modeling and parameters

The primary focus of our analysis is characterizing seasonality in $X_{CO_2}$ and exploring how and why this seasonality differs across northern high latitude regions, with particular emphasis on the Boreal Forest. To this end, time-series are constructed using daily averaged $X_{CO_2}$ from satellite retrievals, ground-based solar-viewing FTIR spectrometers, and model estimates (see previous methods sections for details). Seasonal cycles are characterized following methods used by Lindqvist et al. (2015), in which daily $X_{CO_2}$ are fit to a skewed sine wave of the form

$$f(t) = a_0 + a_1 t + a_2 \sin\left(\omega[t - a_3] + \cos^{-1}[a_4 \cos(\omega[t - a_5])]\right) \tag{1}$$

where $t$ is days, $\omega = \frac{2\pi}{365.25}$, the interannual trend is defined by $a_0 + a_1 t$, and the seasonal cycle amplitude (SCA) is defined by $2|a_2|$. As a metric for seasonal timing we define half drawdown day (HDD) as the day of year when the detrended seasonal cycle fit, $f(t) - a_0 - a_1 t$, crosses zero from positive to negative. The fit is calculated using nonlinear least squares optimization with the standard error, $\sigma$ defined as the mean of daily standard deviations in $X_{CO_2}$ over all days in the time-series. If there is no daily standard deviation, as is the case with single point model estimates near ground sites from the CT2019B and GC-CT2019 models, standard error is assumed to be 0.25 ppm. Specifically, we implemented a least squares fitting algorithm (we

use the Levenberg-Marquardt algorithm for improved convergence behavior), which seeks to minimize

$$\chi^2 = \sum_{i=1}^{n} \frac{[y_i - f(x_i, a_o, a_1, ... a_j)]^2}{\sigma^2},$$

(2)

and yields variance for each parameter in the fit equation given by

$$\sigma_{a_j}^2 = ([\mathbf{J}^T \mathbf{W} \mathbf{J}]^{-1})_{jj}$$

(3)

where $\mathbf{J}$ is the Jacobian and $\mathbf{W} = \mathbf{V}_{x_i}^{-1}$, the inverse of the covariance matrix. In this case, the variances are scaled by $\chi^2$, so that $\mathbf{V}_{x_i}$ is defined as $\chi^2 \sigma^2 \mathbf{I}$. The variance in fit parameters are taken to be a direct estimate of fit uncertainty and translate to uncertainty in SCA, defined as $2\sigma_{a_2}$ and depicted in figures as errorbars, where appropriate.

The seasonal cycle fitting approach used by Lindqvist et al. (2015) was found to be more numerically stable than fitting to a truncated Fourier Series, as has been employed in previous studies (Wunch et al., 2013; Thoning et al., 1989), because
periods of missing data can produce unrealistic oscillations in a fit to a truncated Fourier series. Even in cases with continuous data, the fit to a truncated Fourier series can yield unrealistic oscillations within the overall seasonal cycle that do not appear to accurately depict variability in the data. These unrealistic oscillations are more pronounced when there are gaps in the time-series of data. The fits to Eq. 1 also show a degradation of the seasonal cycle shape with larger gaps in the time-series, but yield more stability in fitting time-series with some data gaps than the fits to a truncated Fourier series. This is an important
consideration for high latitude regions, which have winter gaps in observations for most passive atmospheric remote sensing measurements, due to lower solar elevation angles or night at satellite overpass time (near solar noon).

## 2.5  CAMS model estimates

Model estimates of $X_{CO_2}$ from the Copernicus Atmospheric Monitoring Services (CAMS) global inversion-optimized greenhouse gas flux model v19r1 are used here as a model comparison to OCO-2 and NNG data. The modeling framework for
CAMS $CO_2$ flux inversions is described in detail by Chevallier (2020b). Quality assessments for the Northern Hemisphere by Chevallier (2020a) report that nearly all biases in both CAMS estimates of in situ $CO_2$ relative to unassimilated aircraft observations and CAMS estimates of $X_{CO_2}$ relative to TCCON observations are within 1 ppm, with standard deviation in bias around 2 ppm. CAMS estimates of $X_{CO_2}$ are available as 3-hourly estimates with 1.9° latitude by 3.75° longitude spatial resolution, which is sufficient for providing multiple grid-points within each zone and coincidence with most ground sites
within approximately 100 km (see Table S1 in the supplement for exact coordinates of grid-points nearest to the ground sites). Daily averages and standard deviation in CAMS $X_{CO_2}$, used to calculate seasonal cycle fits are taken from combined spatial aggregations within zones or coincidence regions and temporal aggregations for each 24 hour date in UTC. We use CAMS model estimates with data assimilation from a global network of surface in situ observations at 119 locations, but without any satellite data assimilation. In addition to the $X_{CO_2}$ estimates, the CAMS model output includes surface flux estimates, which
will be considered further in the Discussion, Sect. 4.2. Both CAMS flux estimates and CAMS $X_{CO_2}$ estimates use the same atmospheric transport modeling framework.

## 2.6 CarbonTracker2019 model estimates

The CarbonTracker2019 (CT2019) model is an inverse model that provides estimates of global $CO_2$ fluxes with a 1° by 1° spatial resolution and estimates of global $X_{CO_2}$ fields with a atmospheric transport simulated by the TM5 model (Krol et al., 2005) with a spatial resolution of 2° latitude by 3° longitude (Jacobson et al., 2020a, b). The model assimilates in situ measurements of atmospheric $CO_2$ concentrations from aircraft, AirCore (Karion et al., 2010), tall tower, and surface measurement platforms at 460 sites around the world. For CT2019 fluxes considered in this analysis and used in GEOS-Chem simulations we use the original CT2019 release from May 2020 (Jacobson et al., 2020a), but for the posterior estimates of $X_{CO_2}$ we use results from an updated release, CT2019B (Jacobson et al., 2020b). This was a matter of practical necessity because of the timing of the updated CT2019B release and the fact that $X_{CO_2}$ estimates from the previous CT2019 version were unavailable after this release. The purpose of the release of CT2019B was reported as, "Correction of a minor bug in CT2019" (Jacobson et al., 2020b). CT2019 uses the Global Fire Emissions Database, Version 4.1 (GFED4.1s; Randerson et al., 2018; van der Werf et al., 2017) as one part of the fire module that estimates emissions from biomass burning, while fossil fuel emissions are driven by a combination of the Miller emissions inventory (see Jacobson et al. (2020b) for more details) and the Open-Source Data Inventory for Anthropogenic Carbon Dioxide (ODIAC; Oda and Maksyutov, 2011) emissions datasets. Biospheric exchange is driven by the Carnegie-Ames Stanford Approach (CASA) biogeochemical model introduced by Potter et al. (1993). In this analysis we calculate seasonal cycle fits using daily CT2019B posterior $X_{CO_2}$ and compare these to corresponding seasonal cycle fits from ground-based and satellite-based observations. The CT2019 estimates of biospheric $CO_2$ exchange are also used within a GEOS-Chem model framework, described in Sect. 2.7, and considered in an assessment of the role of fluxes in the shaping $X_{CO_2}$ seasonality.

## 2.7 GEOS-Chem CO$_2$ and Transport Tracer simulations

The GEOS-Chem atmospheric transport model version 12.7.2 (more detailed information at www.geoschem.org) has 2° latitude by 2.5° longitude spatial resolution, using MERRA-2 meteorology (Gelaro et al., 2017). We use the GEOS-Chem $CO_2$ simulation (Nassar et al., 2010) and GEOS-Chem surface contact tracers, for 2014-2016, to examine the relationships between seasonal cycle parameters and atmospheric transport patterns and speculate on the role of atmospheric transport in determining spatial distributions of $X_{CO_2}$ seasonality across northern high latitudes. The GEOS-Chem $CO_2$ simulation provides daily $X_{CO_2}$ estimates and source attributions for total column $CO_2$, with biospheric fluxes for land and ocean taken from the CT2019 model (Jacobson et al., 2020a), so we refer to this combination of GEOS-Chem atmospheric transport and CT2019 biospheric exchange as GC-CT2019 throughout this paper. Similar to CT2019, this GC-CT2019 simulation uses GFED4.1 to estimate emissions from biomass burning, but unlike CT2019, it uses the Community Emissions Data System (CEDS; Hoesly et al., 2018) for fossil fuel emissions and results from Bukosa et al. (2021) for the chemical production of $CO_2$ in the atmosphere. For the fossil fuel emissions, the CEDS inventory ended in 2014, so 2015 and 2016 emissions were scaled by the CEDS 2014 emissions to match the global total in those later years, as reported by ODIAC. We used this approach, rather than using ODIAC alone, because the CEDS inventory includes anthropogenic biofuel emissions that are not in ODIAC. The GC-CT2019 simu-

lation is initialized on January 1, 2007 with observed global ocean surface mean $CO_2$ (Dlugokencky and Tans, NOAA/GML, www.esrl.noaa.gov/gmd/ccgg/trends/, accessed 2020-12-15), giving the model 7 years of spinup before generating the output for 2014-2016 used here. GC-CT2019, like CT2019B posterior $X_{CO_2}$ estimates are obtained as daily values for grid points with daily averages and standard deviation calculated spatially for each 5° latitude by 20° longitude zone or 5° latitude by 10°

longitude satellite coincidence region; there is no temporal averaging performed on these estimates for this analysis. Unlike the CAMS model and CT2019B $X_{CO_2}$ estimates, which provide optimized $CO_2$ flux and $X_{CO_2}$ estimates using internally consistent atmospheric transport models, GC-CT2019 uses CT2019 fluxes that are estimated using TM5 to simulate atmospheric transport rather than the GEOS-Chem transport model. In addition, the fossil fuel and biomass burning flux estimates used in GC-CT2019 are based on slightly different source datasets. GEOS-Chem simulations are run for 2014-2016 rather than for

2014-2019, like the OCO-2 observations and other model estimates. Analysis shown in the supplement (see Fig. S39) suggests that the spatial distributions of SCA and HDD across northern high latitude zones are not likely to significantly change for CAMS, CT2019B, or OCO-2 when calculated for 2014-2016 instead of 2014-2019. In particular, changes in SCA for the different time periods are less than approximately 0.5 ppm, which represents less than 10% of the $\sim 5$ ppm variability in SCA across these northern high latitude regions. There are some larger discrepancies between OCO-2 HDD for 2014-2016 and

OCO-2 HDD for 2014-2019, with a couple of zones yielding a difference of around 8 days, but this may be partially attributed to the fact that some zones lack sufficient data points in the 2014-2016 time period for a stable and accurate seasonal phase determination.

To examine the role of atmospheric transport in shaping the seasonal cycle of $X_{CO_2}$, we define new tracers of airmass surface contact in the GEOS-Chem atmospheric transport model. These surface contact tracers are emitted uniformly and continuously

over specific surface types at a rate of 1 molecule m$^{-2}$ s$^{-1}$, transported like $CO_2$ and other constituents, and decay with a prescribed e-fold lifetime. One set of these surface contact tracers is emitted uniformly over the ocean and another set is emitted uniformly over land. For both surfaces, we release multiple tracers with lifetimes of 5, 15, 30, and 90 days, making 8 surface contact tracers in total. After spinning up the simulation for several e-fold lifetimes, the tracer concentration at a point in the model indicates the integrated upwind contact with land or ocean over the time scale of the tracer lifetime. For

example, high concentrations of the land surface contact tracers reveal locations where atmospheric circulations have confined air over continents. These surface contact tracers are like e90 Prather et al. (2011), except that e90 is emitted uniformly from all surfaces and therefore indicates upwind contact with any surface rather than particular surface types as we have done. The sum of our land and ocean tracers with 90-day lifetimes is equal to e90. The surface contact tracers are initialized and run for the same periods as the GC-CT2019 simulation, 2014-2016. The surface contact tracers for a given zone were found to vary

minimally in time, so a total zonal average of surface contact tracer contributions was taken spatially and temporally within each zone (see map in Fig. 1) and over all days in 2014-2016.

## 2.8 Treatment of $CO_2$ flux estimates

The role of fluxes in determining $X_{CO_2}$ seasonality at northern high latitudes is assessed using flux estimates from CAMS, as well as the sum of $CO_2$ flux estimates from the CEDS and GFED4.1s inventories (Hoesly et al., 2018; Randerson et al.,

2018; van der Werf et al., 2017) and CarbonTracker2019 biospheric fluxes from land and ocean (Jacobson et al., 2020a) used to generate the GC-CT2019 $X_{CO_2}$ estimates. While the CAMS v19r1 and CT2019B model frameworks have internally consistent atmospheric transport because both $CO_2$ flux and $X_{CO_2}$ estimates are generated using the same atmospheric transport model (Chevallier, 2020b; Jacobson et al., 2020a), GC-CT2019 includes biospheric fluxes from CarbonTracker2019 using the TM5

transport model, but then applies GEOS-Chem atmospheric transport modeling to estimate $X_{CO_2}$. In addition, the fossil fuel fluxes from CEDS and biomass burning fluxes from GFED4.1s may include other assumptions about atmospheric transport.

First, $CO_2$ fluxes from each model are averaged spatially within each 5° latitude by 20° longitude zone (see Fig. 1) for each 3-hourly time-step. A total average annual flux is calculated for each zone by summing all 3-hourly $CO_2$ fluxes in each year and taking an average over the six years in 2014-2019. To calculate the flux seasonal cycle amplitude (flux SCA), the 3-hourly,

spatially-averaged fluxes within each zone are summed for each 24-hour period in UTC and used to derive a 15-day rolling mean, which is then averaged by day of year to yield an average annual cycle for 2014-2019. The difference between the maximum and minimum of the average annual cycle is then taken to be the flux SCA. The annual cycles of fluxes are plotted in Figures S32 through S49 of the supplemental materials.

## 3 Results

### 3.1 $X_{CO_2}$ seasonal cycles near ground sites

Before attempting to interpret spatial distributions of seasonal cycle parameters on continental scales, it is of value to get a better idea of how seasonal cycle parameters from observations at a single location compare to those from spatially averaged data. To this end, five high latitude sites are considered with long term ground-based observations, as described in Sect. 2.2. There are three spatial scales considered with seasonal cycle fits to near noon ground-based (NNG) observations and seasonal

cycle fits to spatially averaged data over a commonly used satellite coincidence region of 5° latitude by 10° longitude centered on the location of the ground site, as well as over the 5° latitude by 20° longitude zone in which the ground site is located (reference zones in Fig. 1). In Fig. 2, observed SCA and HDD from NNG and OCO-2 are correlated against model-derived SCA and HDD from CAMS, CT2019B, and GC-CT2019 at each of the three spatial scales, and the corresponding linear regression equations and correlation coefficients are reported in Table 2. These correlations exhibit tight linearity for SCA (with most

$R^2 > 0.7$) and reasonable linearity for HDD when comparing observed and model-derived parameters at all scales. SCA from CAMS or CT2019B are in better agreement with observations than those from GC-CT2019 in every case, as demonstrated by the fact that the CAMS and CT2019B linear regressions fall closer to the $y = x$ line than the GC-CT2019 linear regression in every panels (a), (b), and (c) Fig. 2. Agreement between model-derived and observed HDD is better for the single-point model estimates nearest the ground site versus NNG results in panel (d) of Fig. 2, and the scatter increases in the comparisons of

HDD from spatially averaged model estimates versus spatially averaged OCO-2 observations in panels (e) and (f) of Fig. 2. All three models tend to yield slightly larger SCA than observations at all sites, as well as later HDD than observations with the exception of CAMS HDD at Bremen and Sodankylä. These results stand in contrast to earlier work by Yang et al. (2007) who found that the Transcom model underestimated SCA of $CO_2$ mixing ratios relative to aircraft observations at nearly every

**Table 2.** Linear regression equations and correlation coefficients for the correlations, presented in Fig. 2, of model-derived versus observed SCA and HDD at three spatial scales for five ground sites.

| Panel | CAMS fit | CT2019B fit | GC-CT2019 fit |
|-------|----------|-------------|---------------|
| (a) | y=0.84x+1.92, $R^2 = 0.840$ | y=0.80x+2.46, $R^2 = 0.919$ | y=0.90x+1.98, $R^2 = 0.872$ |
| (b) | y=0.84x+1.94, $R^2 = 0.657$ | y=0.64x+4.04, $R^2 = 0.597$ | y=1.04x+0.87, $R^2 = 0.754$ |
| (c) | y=0.89x+1.31, $R^2 = 0.797$ | y=0.67x+3.66, $R^2 = 0.729$ | y=1.03x+0.91, $R^2 = 0.834$ |
| (d) | y=0.86x+19.63, $R^2 = 0.304$ | y=0.85x+33.17, $R^2 = 0.563$ | y=0.47x+98.21, $R^2 = 0.622$ |
| (e) | y=0.13x+141.06, $R^2 = 0.015$ | y=0.42x+104.04, $R^2 = 0.432$ | y=0.35x+117.36, $R^2 = 0.432$ |
| (f) | y=0.31x+110.00, $R^2 = 0.059$ | y=0.57x+77.95, $R^2 = 0.522$ | y=0.48x+94.97, $R^2 = 0.385$ |

altitude. Details of the full time-series, plots of seasonal cycle fits, and seasonal cycle fit parameters, with uncertainties, for these ground sites, coincidence regions, and encompassing zones are reported in supplemental materials.

In Fig. 3 and Table 3 the relationships between seasonal cycle parameters from spatially averaged data versus those from a single point, at or nearest the ground sites are explored. In this case, NNG are correlated against spatially averaged OCO-2 retrievals, while model estimates near the ground site are correlated against spatially averaged model estimates. Jacobs et al. (2020b) have shown that an alternative bias correction, parameterized for temperature at 700 hPa, resulted in reduced seasonality in OCO-2 bias within the 5° latitude by 10° longitude coincidence region relative to NNG observations at East Trout Lake, Sodankylä, and Fairbanks. Results in the supplement show that the alternative bias correction improved agreement in both SCA and HDD between NNG seasonal cycles and coincident OCO-2 seasonal cycles. The results in Fig. 3 and Table 3 indicate that HDD correlations across scales tend to be slightly weaker and markedly different depending on whether one considers observed or model-derived seasonal cycles. For the coincidence region and the encompassing zone, OCO-2 data consistently yield later HDD than NNG, while spatially averaged model estimates tend to yield HDD that is in good agreement or slightly earlier than the point nearest to the ground site. SCA for ground sites are well correlated and mostly in close agreement with both SCA for 5° latitude by 10° longitude coincidence regions and SCA for encompassing 5° latitude by 20° longitude zones, demonstrating that SCA scales well with spatial averaging and is a credible metric to consider in the context of this analysis. The relatively weaker correlations in HDD across spatial scales in panels (c) and (d) of Fig. 3 suggest greater spatial heterogeneity in HDD within zones from both observed and model-derived seasonal cycles. This result, in combination with the scatter in panels (d), (e), and (f) of Fig. 2, implies that HDD may display more random variability than SCA, and may therefore be a less useful metric in this context.

## 3.2 $X_{CO_2}$ seasonal cycles by zone

The full set of seasonal cycle fit parameters, corresponding uncertainty estimates, as well as plots of time-series and seasonal cycle fits for each zone and ground site in Fig. 1 are reported in the supplement. While the fits to model estimates are generally similar in shape with fits to observations, there are some zones that yield an unrealistic drop in wintertime values in the OCO-

**Table 3.** Linear regression equations and correlation coefficients for the correlations in Fig. 3 in which SCA and HDD from NNG or model estimates near the ground sites are compared to SCA and HDD from spatially averaged satellite observations or spatially averaged model estimates over the corresponding 5° latitude by 10° longitude coincidence regions and encompassing 5° latitude by 20° longitude zones.

| Panel | observed fit | CAMS fit | CT2019B fit | GC-CT2019 fit |
|-------|-------------|----------|-------------|---------------|
| (a) | y=0.82x+1.44, $R^2 = 0.808$ | y=1.03x+-0.47, $R^2 = 0.986$ | y=0.91x+0.82, $R^2 = 0.995$ | y=1.13x+-1.38, $R^2 = 0.984$ |
| (b) | y=0.92x+0.71, $R^2 = 0.889$ | y=1.04x+-0.55, $R^2 = 0.958$ | y=0.86x+1.25, $R^2 = 0.901$ | y=1.12x+-1.21, $R^2 = 0.974$ |
| (c) | y=1.38x+-60.07, $R^2 = 0.666$ | y=1.07x+-12.53, $R^2 = 0.899$ | y=0.94x+9.53, $R^2 = 0.990$ | y=1.50x+-89.00, $R^2 = 1.000$ |
| (d) | y=1.29x+-45.32, $R^2 = 0.698$ | y=1.17x+-28.55, $R^2 = 0.873$ | y=1.01x+-3.56, $R^2 = 0.899$ | y=1.94x+-167.50, $R^2 = 0.958$ |

seasonal cycle fits, which is more pronounced for zones with fewer satellite-based $X_{CO_2}$ observations near the peak and trough of the seasonal cycle. This wintertime drop in the shape of the seasonal cycle is evidence that should motivate further efforts to increase satellite-based observations over high latitude regions outside of the summer months. Only a small number of zones, particularly in the Asian Tundra, have seasonal cycle fits that are obviously compromised by insufficient data in spring and autumn, while the majority of zones have seasonal fits that look reasonable and are similar to seasonal cycle fits to continuous model estimates, with the exception of the noted wintertime drop. An analysis of changes in CAMS seasonal cycle fits when selecting only days with OCO-2 observations available rather than the full continuous time-series of model estimates is presented in the supplement, which indicates that shifts in CAMS SCA due to imposed data gaps are -0.044 $\pm$ 0.197 ppm, with shifts in SCA for all zones less than 0.5 ppm, while biases in CAMS SCA for the full time-series relative to OCO-2 SCA are 0.547 $\pm$ 0.720 ppm and range from -1.0 to 3.0 ppm. These shifts in CAMS SCA are also small relative to the approximately 5 ppm variability in SCA seen across the northern high latitude regions. In addition, the Lindqvist method of fitting (Lindqvist et al., 2015) still provides a more constrained shape than a fit to a truncated Fourier series, which can yield highly unrealistic oscillations in time-series with only minimal gaps in data coverage. The close similarities between spatial distributions of seasonal cycle parameters from OCO-2, CAMS, CT2019B, and GC-CT2019 are apparent in Fig. 4.

### 3.2.1 Comparing observed and model-derived SCA and HDD

Direct correlations of model-derived versus observed SCA and HDD are shown in Fig. 5, indicating that model-derived and observed SCA agree with tight linearity ($R^2 > 0.68$) while model-derived and observed HDD agree with reasonable linearity ($R^2 > 0.45$). Model estimates tend to yield slightly larger SCA and later HDD than OCO-2 and NNG. The most notable discrepancies between observed and model-derived SCA tend to occur in the Asian Tundra and North American Tundra regions for which model estimates yield larger values of SCA that are more homogeneous across zones than observations. SCA and HDD from CAMS and CT2019B model-derived seasonal cycles are in better agreement with observed SCA and HDD than those from GC-CT2019. For SCA in particular, the CAMS and CT2019B seasonal cycles are in very close agreement with OCO-2 seasonal cycles when satellite retrievals are treated with the alternative high latitude quality controls and bias correction described by Jacobs et al. (2020b) (see Sect. 2.1 and Sect. S2). SCA and HDD derived from GC-CT2019 (see panels (e) and

(f) of Fig. 5) cover a broader range of values, particularly over-estimating SCA in the Asian Boreal and Tundra regions and exhibiting more scatter in correlations of model-derived versus observed HDD. GC-CT2019 SCA remain strongly correlated with observed SCA, but the slope of this correlation is steeper than for CAMS and CT2019B model estimates.

## 3.3 Spatial distributions of SCA and HDD

Seasonal cycle amplitudes (SCA) and half drawdown day (HDD) are mapped in Fig. 4, showing results from OCO-2 observations, CAMS model estimates, CT2019B model estimates and GC-CT2019 model estimates of $X_{CO_2}$. Figure 4 demonstrates that both OCO-2 observations and model estimates from CAMS, CT2019B, and GC-CT2019 yield larger SCA and earlier HDD in the Asian Boreal Forest than any other region. The earlier seasonal timing of the Asian Boreal Forest zones is consistent with results from Keppel-Aleks et al. (2012) and Schneising et al. (2011), and these studies also linked earlier drawdown in Asia to larger SCA. Although one would not necessarily expect SCA of surface in situ measurements to match SCA of $X_{CO_2}$, this finding also aligns with the study by Lin et al. (2020), who found that Siberia had the largest SCA in surface $CO_2$ concentrations when normalized for area. The CAMS, CT2019B, and GC-CT2019 results in Fig. 4 exhibit more apparent latitudinal gradients, whereas the results from OCO-2 show more spatial heterogeneity, and this is particularly true for HDD. Seasonal cycles of direct observations are expected to display more heterogeneity than seasonal cycles of model estimates, which depend on mathematical modeling of atmospheric transport to calculate $X_{CO_2}$, even if the underlying fluxes are based on in situ data assimilation. Overall, the spatial distributions in SCA and HDD from OCO-2 agree more with those from CAMS and CT2019B than those from GC-CT2019. GC-CT2019 yields larger magnitudes of SCA for many regions, as well as SCA and HDD spread across a larger range for northern high latitude regions, relative to SCA from OCO-2, CAMS, or CT2019B. In addition, OCO-2 observations yield notably smaller SCA than CAMS, CT2019B, or GC-CT2019 in the western zones of the Asian Boreal Forest, in the Asian Tundra, and in the eastern zones of North America.

Panels (a), (c), (e), and (g) of Fig. 6 show a clear increase in SCA from west to east across the Eurasian continent in both model-derived and observational results. In North America, longitudinal gradients are more subtle. While OCO-2 observations exhibit a slight gradient in SCA across North America that increases east to west, CAMS, CT2019, and GC-CT2019 yield SCA that increase from west to east, in the same direction as gradients across Eurasia. This discrepancy in North America hinges primarily on the zones of Boreal Forest and Tundra in eastern North America, which have the smallest SCA for that continent when using OCO-2 data, but have the largest SCA for that continent when using CAMS, CT2019B, or GC-CT2019 model estimates. As expected, panels (a), (c), (e), and (g) in Fig. 6 show latitudinal gradients with increasing SCA from south to north for both observed and model-derived seasonal cycles. However, the Asian Boreal Forest zones stand apart from other regions in all panels of Fig. 6, particularly when plotted against latitude, with larger SCA than other data at similar latitudes or longitudes. Results in Fig. 7 demonstrate that HDD are far more scattered and do not follow the distinct trends with latitude and longitude that SCA does. HDD spatial gradients seem to be inverted relative to SCA with a vague tendency toward later HDD at more northern latitudes and more western longitudes, and HDD also exhibits similar discrepancies between observed and model-derived longitudinal gradients for North America.

**Table 4.** Linear regression equations and correlation coefficients for the correlations in Fig. 8 in which SCA is correlated against HDD for observed and model-derived results. Linear regressions were calculated considering all 5° latitude by 20° longitude zones, and then separately for zones in temperate, boreal, and tundra regions.

| Ecoregion(s) | observed fit | CAMS fit | CT2019B fit | GC-CT2019 fit |
|---|---|---|---|---|
| All | y = -0.11x + 28.75, $R^2$ = 0.193 | y = -0.03x + 15.93, $R^2$ = 0.017 | y = -0.14x + 34.65, $R^2$ = 0.302 | y = -0.18x + 42.75, $R^2$ = 0.373 |
| Temperate | y = -0.10x + 27.01, $R^2$ = 0.151 | y = -0.05x + 17.77, $R^2$ = 0.077 | y = -0.16x + 37.42, $R^2$ = 0.556 | y = -0.15x + 37.05, $R^2$ = 0.606 |
| Boreal | y = -0.18x + 41.54, $R^2$ = 0.514 | y = -0.15x + 37.26, $R^2$ = 0.463 | y = -0.21x + 48.17, $R^2$ = 0.815 | y = -0.24x + 54.27, $R^2$ = 0.843 |
| Tundra | y = -0.21x + 46.85, $R^2$ = 0.751 | y = -0.14x + 36.45, $R^2$ = 0.724 | y = -0.15x + 38.25, $R^2$ = 0.679 | y = -0.20x + 48.05, $R^2$ = 0.878 |

### 3.4 The relationship between SCA and HDD

Figure 8 and the calculated linear regressions in Table 4 show that there is a negative correlation between HDD and SCA for both observed and model-derived seasonal cycle fits, such that earlier HDD corresponds to larger SCA. CAMS model estimates yield correlations between HDD and SCA that are more similar to those from OCO-2 and NNG measurements, while GC-CT2019 yields stronger correlations with steeper slopes. The results in Fig. 8 emphasize the exceptionally early HDD and large SCA of the Asian Boreal Forest, such that many of the Asian Boreal zones fall more in line with the tundra zones than with the other boreal zones. A latitudinal gradient is suggested by the fact that the linear regressions plotted in Fig. 8 are shifted up for the tundra zones and shifted down for the temperate zones, with the boreal zones in between the two. Furthermore, the separate linear regressions for the temperate, boreal, and tundra zones plotted in Fig. 8 have much larger $R^2$ than the linear regressions for all the zones together (see Table 4), suggesting that there are different dynamics in different biomes that affect relationships between extent and timing of apparent $CO_2$ uptake. The strength of this correlation in observed and CAMS seasonal cycles was highest for tundra, and lowest for temperate zones with the boreal zones falling in between.

### 3.5 Comparing GEOS-Chem surface contact tracers to observed SCA and HDD

GEOS-Chem surface contact tracers were used to simulate the release of tracers from land and ocean yielding relative concentrations of tracers with lifetimes of 5, 15, 30, and 90 days for a given grid point and day. In these results the relative contributions of surface contact tracers for each zone represent an overall average of surface contact tracer contributions for all days in 2014-2016, and aggregated spatially across the 5° latitude by 20° longitude zone. The surface contact tracers show that the largest $X_{CO_2}$ SCA occurs in areas with the greatest influence from air that contacted land surfaces 15 to 30 days prior. There are clear similarities in the spatial distributions of SCA in panels (a), (c), and (e) of Fig. 4 and those of the surface contact tracers from land with a 15 or 30 day lifetime, as shown in panels (c) and (e) of Fig. 9. There are also similarities in the spatial distributions of HDD in panels (b), (d), and (f) of Fig. 4 and those for the surface contact tracers from ocean with a 15 or 30 day lifetime, as shown in panels (d) and (f) of Fig. 9. The relative strength of linear relationships between seasonal cycle parameters from OCO-2 observations and surface contact tracers are quantified with correlation coefficients in Fig. 10.

Figure 10 shows that the observed SCA are most correlated with land-based surface contact tracers with 15 day and 30 day atmospheric lifetimes, while observed HDD are most correlated with ocean-based surface contact tracers with 15 day and 30 day atmospheric lifetimes. Correlations between HDD and land tracers are weak (see supplement Fig. S45) and seem to follow a curve rather than a line, or may be representative of two different linear relationships for different groups of zones. The correlations with ocean tracers were always inverted relative to those with land tracers, such that reduced contribution from ocean tracers and increased contributions from land tracers consistently correspond with larger SCA and often correspond with earlier HDD.

## 4   Discussion

In this analysis, methods described by Lindqvist et al. (2015) were used to fit daily average time-series of daily $X_{CO_2}$ to a skewed sine wave (see Eq. 1) and subsequently calculate seasonal cycle amplitude (SCA) and half drawdown day (HDD), as described in Sect. 2.4. These fitting methods have been found to yield more stable and realistic fits for time-series with winter gaps than fitting to a truncated Fourier Series. Increased OCO-2 throughput (Kiel et al., 2019; O'Dell et al., 2018; Osterman et al., 2018) and use of a bias correction and quality control methods tailored to northern high latitudes (Jacobs et al., 2020b) improve the availability of OCO-2 data at the edges of the growing season and assist in generating stable and realistic seasonal cycle fits. Results in Fig. 3 and Table 3 indicate close agreement between SCA from NNG observations at five ground sites and corresponding SCA from OCO-2 data in the $5°$ latitude by $10°$ longitude spatial coincidence regions. Relatively weak correlations in HDD at the five ground sites, both across spatial scales and between observed and model-derived seasonal cycles, are likely to be at least partly attributable to spatial heterogeneity in seasonal cycle timing within $5°$ latitude by $20°$ longitude zones. It is possible that HDD as a phase metric is not as well constrained by the seasonal cycle fitting methods used here as SCA. Many of the greatest discrepancies in HDD when comparing coincidence regions or zones to ground sites (see Fig. 3 and Table 3) occur in observed seasonal cycles and may arise from disagreement between NNG and OCO-2 observations. Discrepancies between observed and model-derived HDD, which are most apparent for GC-CT2019 $X_{CO_2}$ estimates, could reflect an inaccurate representation of ecosystem respiration in the CASA terrestrial biosphere model, which underlies the CarbonTracker2019 biospheric fluxes used in the GC-CT2019. Byrne et al. (2018) found that differences in the seasonal timing of NEE maximum drawdown were primarily driven by differences in the timing of ecosystem respiration in spring and fall, while the amplitude of NEE was largely influenced by the magnitude of peak gross primary production (GPP). This could explain the higher degree of spatial heterogeneity in seasonal cycle timing because ecosystem respiration is driven by soil temperature, soil moisture, and litter accumulation, which could all be expected to exhibit a higher degree of spatial heterogeneity than ambient temperature and sunlight,. If there is, in truth, more spatial heterogeneity in the timing of seasonal $CO_2$ uptake, then a failure to accurately represent these spatial distributions in timing could be exacerbated by errors or differences in atmospheric transport modeling and result in larger or more variable discrepancies between observed and model-derived HDD. In this case, the relatively spatially homogeneous values of SCA would be easier to accurately predict in the models than HDD. Alternatively, if SCA is primarily defined by the magnitude of maximum GPP, it may be better

constrained in the models because data products like SIF and NDVI can be used to validate model estimates of GPP, while ecosystem respiration does not have a direct data proxy.

OCO-2, CAMS, CT2019B, and GC-CT2019 all yield very similar spatial distributions of SCA and HDD across the northern high latitude regions (see map in Fig. 1), and both had large SCA and early HDD in the Asian Boreal Forest as well as a

clear increase in SCA from west to east across the Eurasian continent. We found an inverse linear relationship between SCA and HDD across northern high latitudes for both observed and model-derived seasonal cycle fits (see Fig. 8 and Table 4), and these correlations were stronger when separating by ecological type (Temperate, Boreal, or Tundra). This relationship is an interesting result of this analysis that warrants a more in depth exploration in future work, though we cannot speculate on the cause of this phenomenon in the context of this study. Discrepancies between SCA from GC-CT2019 and SCA from CAMS,

CT2019B, and observations are consistent with the assessments of seasonal bias resulting from the GEOS-Chem transport modeling framework with MERRA-2 meteorology, as described by Schuh et al. (2019), but may also be partly explained by the lack of internally consistent atmospheric transport modeling for $CO_2$ flux and $X_{CO_2}$ estimates in the GC-CT2019 framework. Schuh et al. (2019) found that the GEOS-Chem $CO_2$ simulation overestimates $X_{CO_2}$ in winter and underestimates $X_{CO_2}$ in summer relative to the TM5 transport model, yielding an exaggerated seasonal oscillation and larger SCA. Byrne et al. (2018)

found that CarbonTracker2016, using CASA to constrain biospheric carbon exchange, estimated later NEE drawdown than flux inversions with either GOSAT or TCCON data assimilation, and this appears to be consistent with the later HDD estimated by the GC-CT2019. Despite these discrepancies, we contend that the strong correlations between GC-CT2019 and observational results (see panels (e) and (f) for Fig. 5) suggest that GEOS-Chem simulations remain a useful tool for investigating the broader implications of spatial distributions in SCA and HDD from OCO-2 observations.

Two limiting hypotheses for the origin of the spatial patterns in $X_{CO_2}$ SCA shown in Fig. 4 are that they arise from differences in flux magnitudes within the 5° latitude by 20° longitude zone or that they arise from transport patterns accumulating $CO_2$ exchanges across multiple zones or regions. To investigate the relative influences of atmospheric transport or fluxes within zones on $X_{CO_2}$ seasonal cycles, we consider source apportionment from the GEOS-Chem $CO_2$ simulation (Nassar et al., 2010) and GEOS-Chem surface contact tracers, as well as surface $CO_2$ flux estimates from CarbonTracker2019 and CAMS models.

## 4.1 The role of atmospheric transport in shaping $X_{CO_2}$ seasonality

Fisher et al. (2010) and Wang et al. (2011) demonstrated the ability of GEOS-Chem to represent synoptic transport from mid-latitudes to the Arctic. Schuh et al. (2019) evaluated meridional transport of $CO_2$ and $SF_6$ in GEOS-Chem, using the same MERRA-2 meteorology used here, and suggested that the model may underestimate vertical and meridional mixing, which slightly increases $CO_2$ seasonal cycle amplitude in GEOS-Chem in high northern latitudes compared to some other

models. Despite some discrepancies in SCA and HDD from GC-CT2019, relative to observed SCA and HDD, which may arise from differences between the GEOS-Chem and TM5 atmospheric transport models, the GEOS-Chem transport model reproduces surface temperature and pressure patterns and thus is a reasonable representation of atmospheric transport (Wang et al., 2011; Fisher et al., 2010). Therefore, comparing these GEOS-Chem surface contact tracers with observed SCA and HDD should provide useful insights into the influence of atmospheric transport patterns on observed $X_{CO_2}$ seasonality. The higher

correlation coefficients obtained when comparing OCO-2 SCA to land and ocean tracers with 15 day and 30 day lifetimes suggest that accumulation of $CO_2$ flux due to atmospheric transport on roughly monthly timescales plays an important role in affecting $X_{CO_2}$ SCA.

## 4.2 The role of $CO_2$ fluxes in $X_{CO_2}$ SCA

Figure 11 shows maps of average annual fluxes and flux SCA for GC-CT2019 and CAMS, which show that flux SCA in panels (c) and (d) are distributed without the apparent longitudinal gradient seen in $X_{CO_2}$ SCA in panels (a), (c), (e), and (g) of Fig. 4. The weak correlations ($R^2 < 0.2$) between flux SCA and $X_{CO_2}$ SCA, shown in panels (a), (c), (e), and (g) of Fig. 12 and quantified in Fig. 13, combined with the relatively strong correlations ($R^2 > 0.6$) between $X_{CO_2}$ and surface contract tracers with 15 and 30 day lifetimes, suggest that accumulation of $CO_2$ flux due to atmospheric transport on roughly monthly

timescales is more influential in determining $X_{CO_2}$ SCA than fluxes within a $5°$ latitude by $20°$ longitude zone. However, there are slightly stronger correlations between average annual flux and $X_{CO_2}$ SCA, shown in panels (b), (d), (f), and (h) of Fig. 12 and quantified in Fig. 13, which suggest some possible link between $X_{CO_2}$ SCA and the relative source or sink strength of a given zone. Panels (a) and (b) of Fig. 11 show that both models predict large negative average annual fluxes for Asian Boreal zones, designating the Asian Boreal region as a major sink for $CO_2$, and suggesting that anomalously large $X_{CO_2}$ SCA in the

Asian Boreal region may be partially influenced by enhancements in $CO_2$ uptake within that region. In another instance, the European Temperate zone 3 and Bremen TCCON site both have exceptionally large positive average annual $CO_2$ flux due to a significant contribution from fossil fuel emissions (as shown in Sect. S5), but they did not yield anomalously large flux SCA because the fossil fuel emissions do not exhibit significant seasonal variability. The European Temperate zone 3 and Bremen also yielded smaller $X_{CO_2}$ SCA and later HDD than most of the other zones and ground sites for both observed and model-

derived seasonal cycles. In this case, the large fossil fuel emissions may be indirectly influencing the $X_{CO_2}$ SCA despite the fact that these emissions are not directly contributing to the seasonal variability in atmospheric $CO_2$ in this zone or at this site.

## 5   Conclusions

Satellite-based instruments, such as OCO-2, open the possibility to study $CO_2$ exchange and transport throughout the vast and largely un-instrumented northern high latitudes. Improvements in retrieval and quality control methods for satellite-based

observations of atmospheric $CO_2$ have allowed for a data-driven investigation of $X_{CO_2}$ seasonality over regions, like Siberia, that have previously been largely inaccessible and unobserved. Model-derived $X_{CO_2}$ SCA from CAMS, CT2019B, and GC-CT2019 agree ($R^2 > 0.68$) with observed SCA patterns from around the northern high latitudes (see Fig. 5). Correlations in Fig. 5 show that CAMS and CT2019B have near unity slopes of model predicted SCA as compared to OCO-2, while GC-CT2019 has a higher than unity slope but still strongly correlated. Our results show that the Asian Boreal Forest region is distinct from

other northern high latitude regions with larger seasonal cycle amplitude (SCA) and earlier half drawdown day (HDD) (see Sect. 2.4), and gradients of increasing SCA and earlier HDD span from west to east across the Eurasian continent. Longitudinal gradients in SCA and HDD across the North American continent are more subtle than longitudinal gradients across the Eurasian

continent. Discrepancies between observed (OCO-2) and model-derived (CAMS CT2019B, and GC-CT2019) SCA and HDD in the eastern zones of North America result in opposing longitudinal gradients in SCA and HDD across the North American continent, such that OCO-2 observations yield increasing SCA from east to west, while model estimates yield increasing SCA from west to east. In order to assess the relative influences of the accumulation of $CO_2$ exchanges during atmospheric transport

or the magnitudes of fluxes within 5° latitude by 20° longitude zones, we compare GEOS-Chem surface contact tracers with observed spatial distributions of SCA and HDD from OCO-2. GEOS-Chem surface contact tracers revealed that the largest $X_{CO_2}$ SCA occur in areas with the greatest influence from land tracers with 15 or 30 day lifetimes. The correlations of $X_{CO_2}$ SCA with land contact tracers are stronger than the correlations of observed $X_{CO_2}$ SCA with SCA of $CO_2$ fluxes or with the total annual $CO_2$ flux within a given 5° latitude by 20° longitude zone. This indicates that accumulation of terrestrial $CO_2$ flux

during atmospheric transport on roughly monthly timescales is a major driver of regional variations in $X_{CO_2}$ SCA, which is at least as important in shaping observed $X_{CO_2}$ seasonality as the terrestrial flux magnitudes within zones. However, there is some correlation between the total average annual fluxes used in the GC-CT2019 and $X_{CO_2}$ SCA, and the Asian Boreal region was still determined to have by far the largest negative fluxes of any of the regions in addition to having the largest $X_{CO_2}$ SCA and earliest HDD. Three important insights about the drivers influencing $X_{CO_2}$ seasonality come out of this analysis. First, a

combination of fluxes within zones and the accumulation of $CO_2$ flux during atmospheric transport affects the observed spatial distributions of $X_{CO_2}$ seasonal cycle parameters. Second, a robust understanding of atmospheric transport patterns on roughly monthly timescales is essential for accurate interpretation of $X_{CO_2}$ seasonality for northern high latitudes. Third, seasonality in $X_{CO_2}$ in northern high latitude regions is almost completely dictated by seasonality in the exchange of $CO_2$ with the terrestrial biosphere. In future work, it would be of value to expand this analysis to assess both long-term temporal trends in $X_{CO_2}$

seasonality, as well as interannual anomalies that may result from global weather patterns such as the polar vortex or El Niño.

*Code availability.* TEXT

*Data availability.*

*Code and data availability.* OCO-2 data and quality control parameters used here are taken from OCO-2 Lite files (version 9, "B9"), and quality filtering and bias corrections are applied following Jacobs et al. (2020b), as described in Sect. 2.1. OCO-2 Lite files are produced

by the NASA OCO-2 project at the Jet Propulsion Laboratory, California Institute of Technology, and obtained from the NASA Goddard Earth Science Data and Information Services Center (GES-DISC; https://daac.gsfc.nasa.gov/). TCCON data are available from the TCCON data archive, hosted by CaltechDATA: https://tccondata.org/. EM27/SUN GGG2014 retrievals from Fairbanks, Alaska are available on the Oak Ridge National Laboratory Distributed Active Archive Center (ORNL DAAC): https://doi.org/10.3334/ORNLDAAC/1831 [these data may still be in the process of being published to the ORNL DAAC, but are expected to be available by the time of manuscript acceptance].

Methods used to bias correct EM27/SUN data to TCCON are described in the supplemental materials for Jacobs et al. (2020b). All ground-

based datasets are also cited individually in Sect. 2.2. CAMS optimized flux-inversion model output is available on the Copernicus website: https://ads.atmosphere.copernicus.eu/cdsapp#!/dataset/cams-global-greenhouse-gas-inversion. GEOS-Chem source code is publicly available (https://doi.org/10.5281/zenodo.3701669). Model output analyzed in this work are archived at Zenodo [data in netCDF format will be uploaded after manuscript acceptance].

*Sample availability.* TEXT

*Video supplement.* TEXT

*Author contributions.* Nicole Jacobs composed this manuscript and conducted the analysis under the supervision of William R. Simpson. Kelly A. Graham and Christopher Holmes ran GEOS-Chem simulations and provided essential guidance in interpreting results. Frank Hase, Thomas Blumenstock, Qiansi Tu, Matthias Frey, Manvendra K. Dubey, and Harrison A. Parker all contributed to data collection with the
EM27/SUNs in Fairbanks, including instrument evaluations, maintenance, and establishing long-term operations in Fairbanks. Debra Wunch contributed data from the East Trout Lake TCCON site, as well as a thorough evaluation of the manuscript. Rigel Kivi and Pauli Heikkinen contributed data from the Sodankylä TCCON site. Justus Notholt, Christof Petri, and Thorsten Warneke contributed data from the Białystok and Bremen TCCON sites. All coauthors have provided essential feedback and insights on the content of the manuscript and supplemental materials.

*Competing interests.* The authors declare that they have no conflict of interest.

*Disclaimer.* TEXT

*Acknowledgements.* The Simpson Lab at UAF acknowledges the Alaska Space Grant Graduate Fellowship and OCO Science Team Grant (NNH17ZDA001N-OCO2) for support. K. A. Graham and C. Holmes acknowledge support from the NSF Office of Polar Programs (grant 1602883) and the NASA Earth and Space Science Fellowship (grant 80NSSC17K0361). KIT acknowledges support by ESA via the projects
COCCON-PROCEEDS, COCCON-PROCEEDS II, and FRM4GHG. M. K. Dubey thanks NASA CMS, LANL LDRD and UCOP support for the LANL EM27/SUN deployments. D. Wunch acknowledges CFI, ORF, and NSERC support for the ETL TCCON station.

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

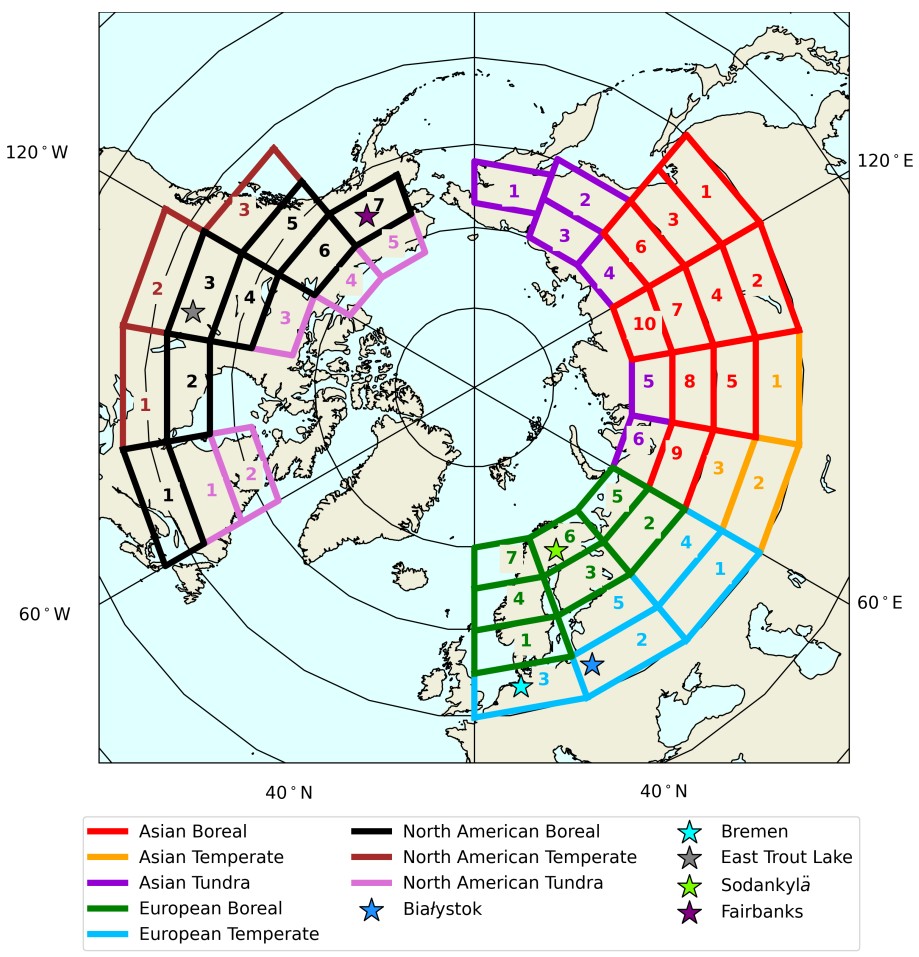

**Figure 1.** Map for of regions, zones, and locations of ground-based $X_{CO_2}$ observations.

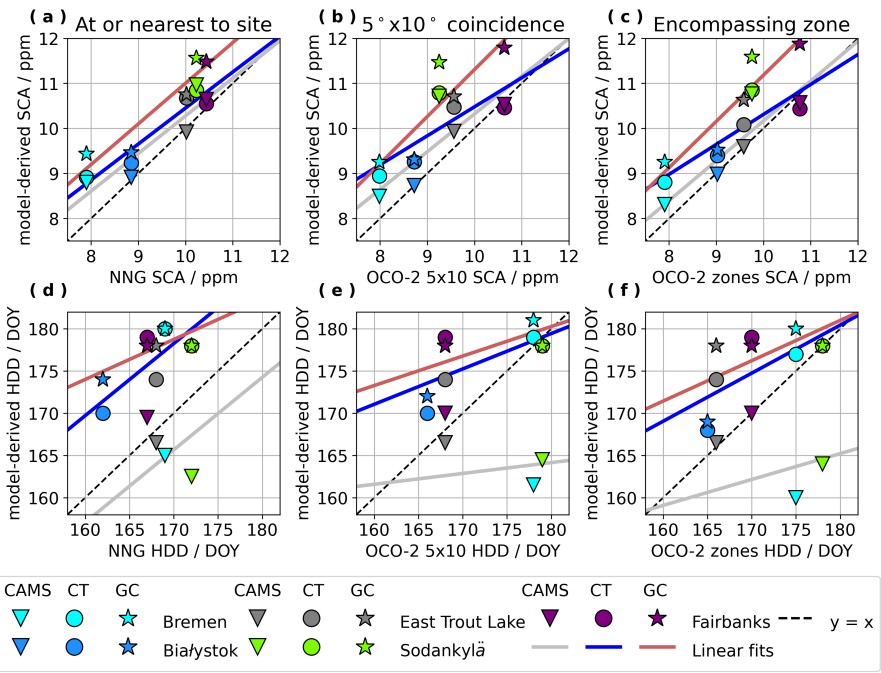

**Figure 2.** Correlations of observed versus model-derived SCA and HDD. Observed $X_{CO_2}$ seasonal cycles are based on NNG and OCO-2 satellite-based observations, while model-derived $X_{CO_2}$ seasonal cycles are based on estimates from the CAMS, CT2019B, and GC-CT2019 model frameworks. SCA and HDD are compared for three spatial types at each of the five ground sites: seasonal fits of near noon ground-based (NNG) observations from TCCON and EM27/SUN measurements are compared to fits of daily model estimates at the nearest model grid-point to the ground-site; seasonal fits of daily average OCO-2 retrievals that fall within the 5° latitude by 10° longitude region of coincidence, centered on the location of each ground site, are compared to fits of daily model estimates averaged across the coincidence region; seasonal fits of OCO-2 daily averages for the 5° latitude by 20° longitude zone containing each ground site are compared to fits of daily model estimates average across those same zones (see map in Fig. 1 and site details in Table 1).

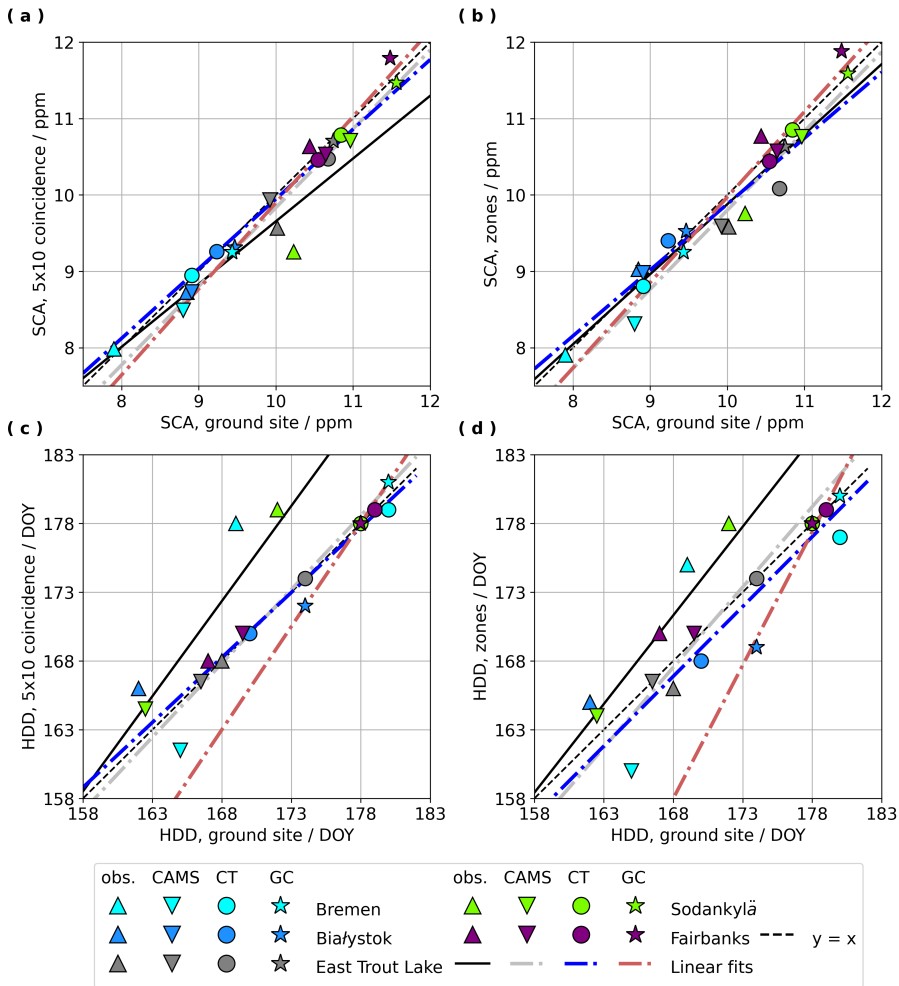

**Figure 3.** Correlations of SCA and HDD from NNG or the model grid-point nearest the ground site versus SCA and HDD from spatially averaged OCO-2 observations or model estimates. For these correlations, we only compare across spatial scales by pairing NNG observations with spatially averaged OCO-2 data and pairing single-point CAMS, CT2019B, and GC-CT2019 model estimates nearest to the ground sites with corresponding spatially averaged model estimates in the same model framework. Note: There are some overlapping points in the correlations of HDD in panels (c) and (d) that may visually obscure some of the results.

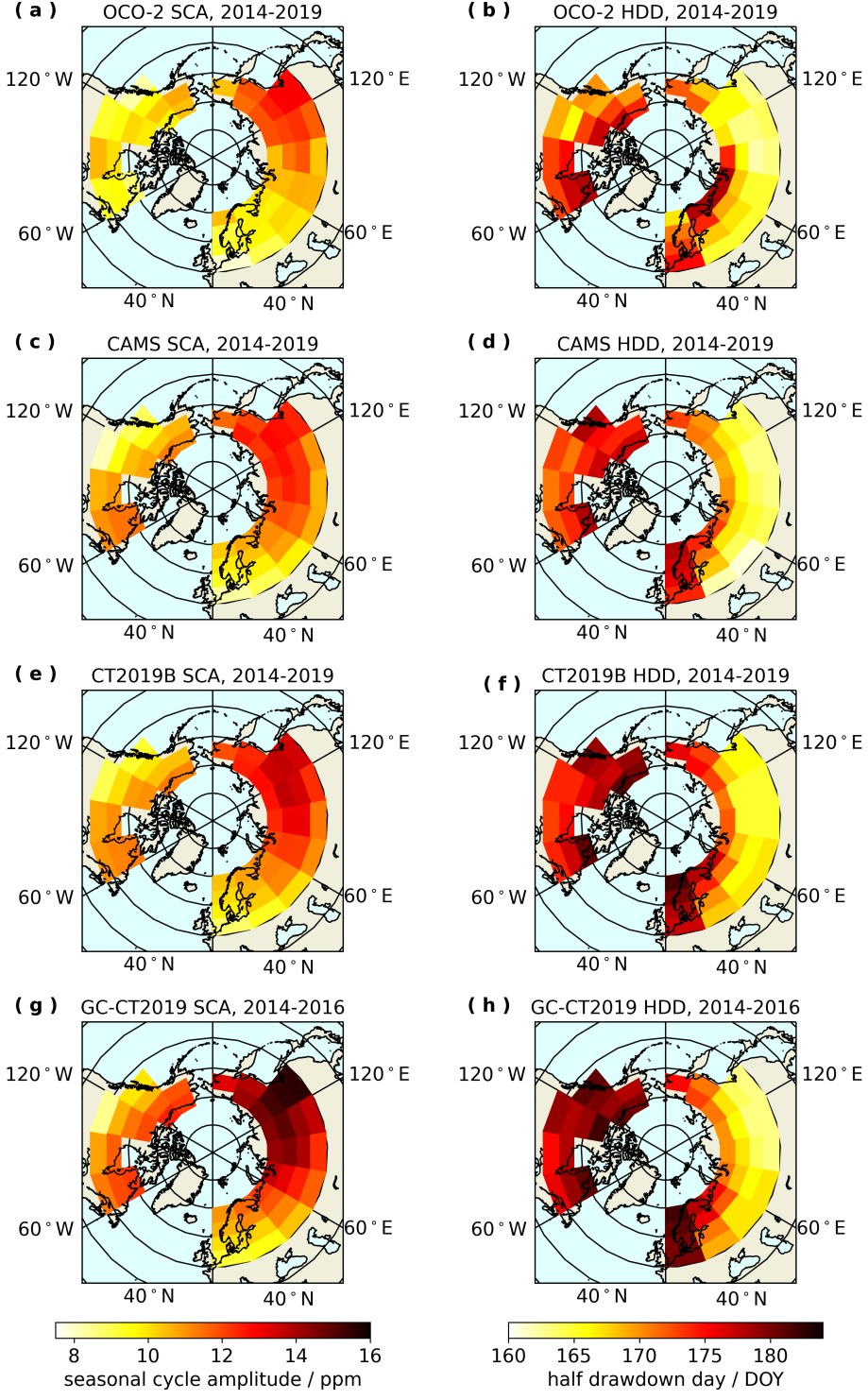

**Figure 4.** Maps of SCA and HDD (represented as color scaling) from seasonal cycle fits to OCO-2 observations and model estimates from CAMS, CT2019B, and GC-CT2019 for each zone defined in Fig. 1.

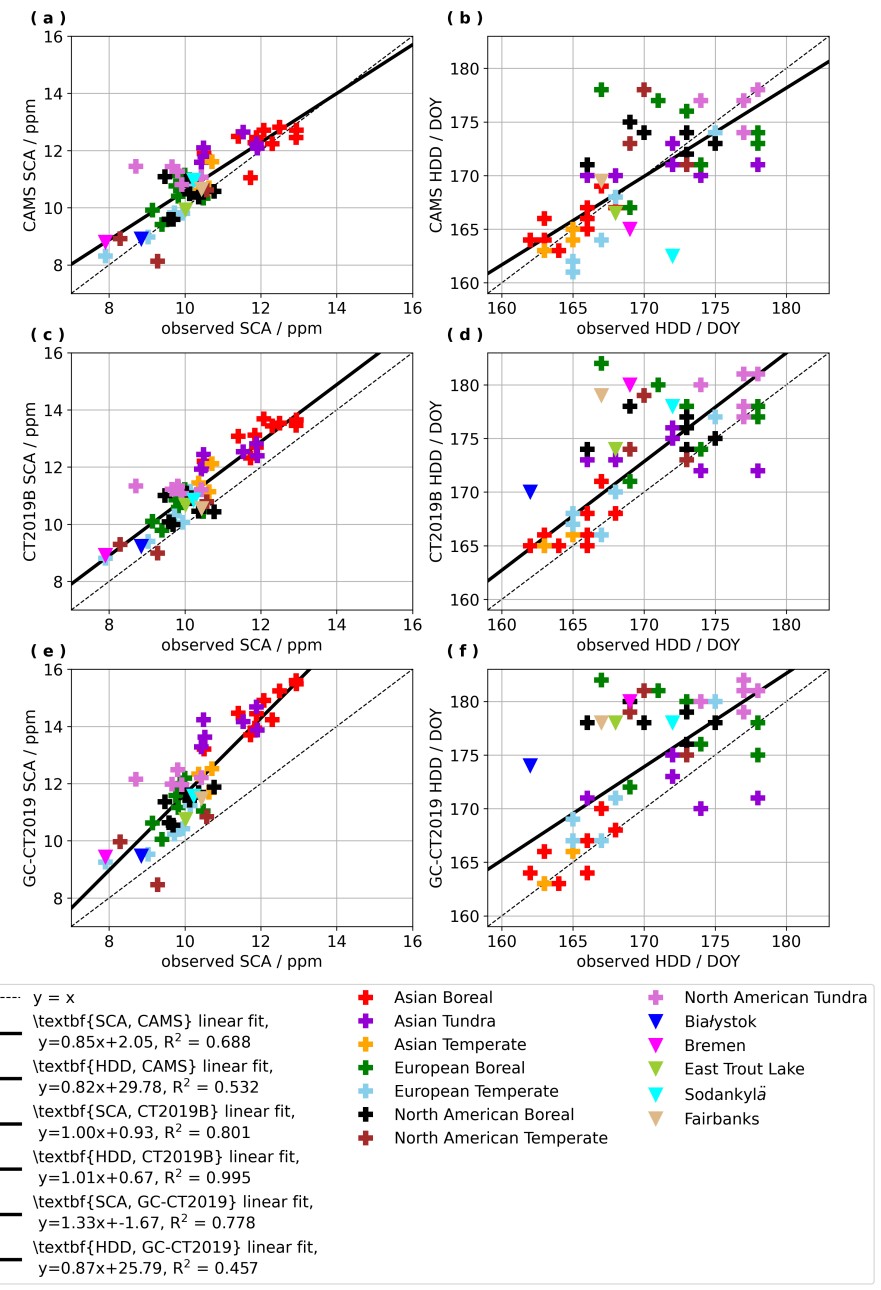

**Figure 5.** Plots of model-derived versus observed SCA and HDD using model estimated from CAMS ((a) and (b)), CT2019B ((c) and (d)), and GC-CT2019 ((e) and (f)), and using observations from OCO-2 within $5°$ latitude by $20°$ longitude zones and NNG at five ground sites.

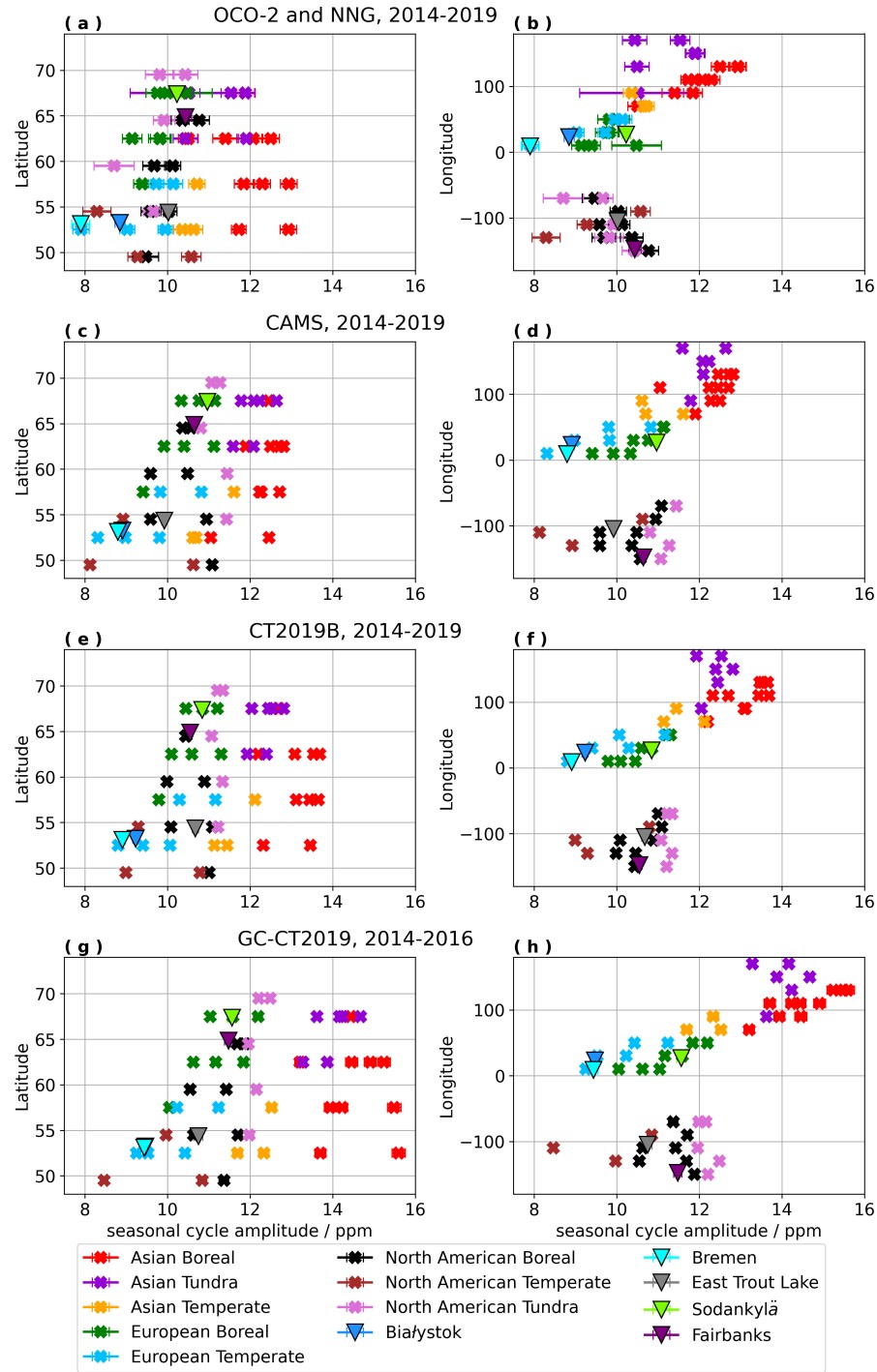

**Figure 6.** Plots of latitude and longitude correlated to SCA using observational results from OCO-2 and NNG observations ((a) and (b)), CAMS model estimates ((c) and (d)), CT2019B model estimates ((e) and (f)), and GC-CT2019 model estimates ((g) and (h)). The latitude and longitude for each zone is located at its center.

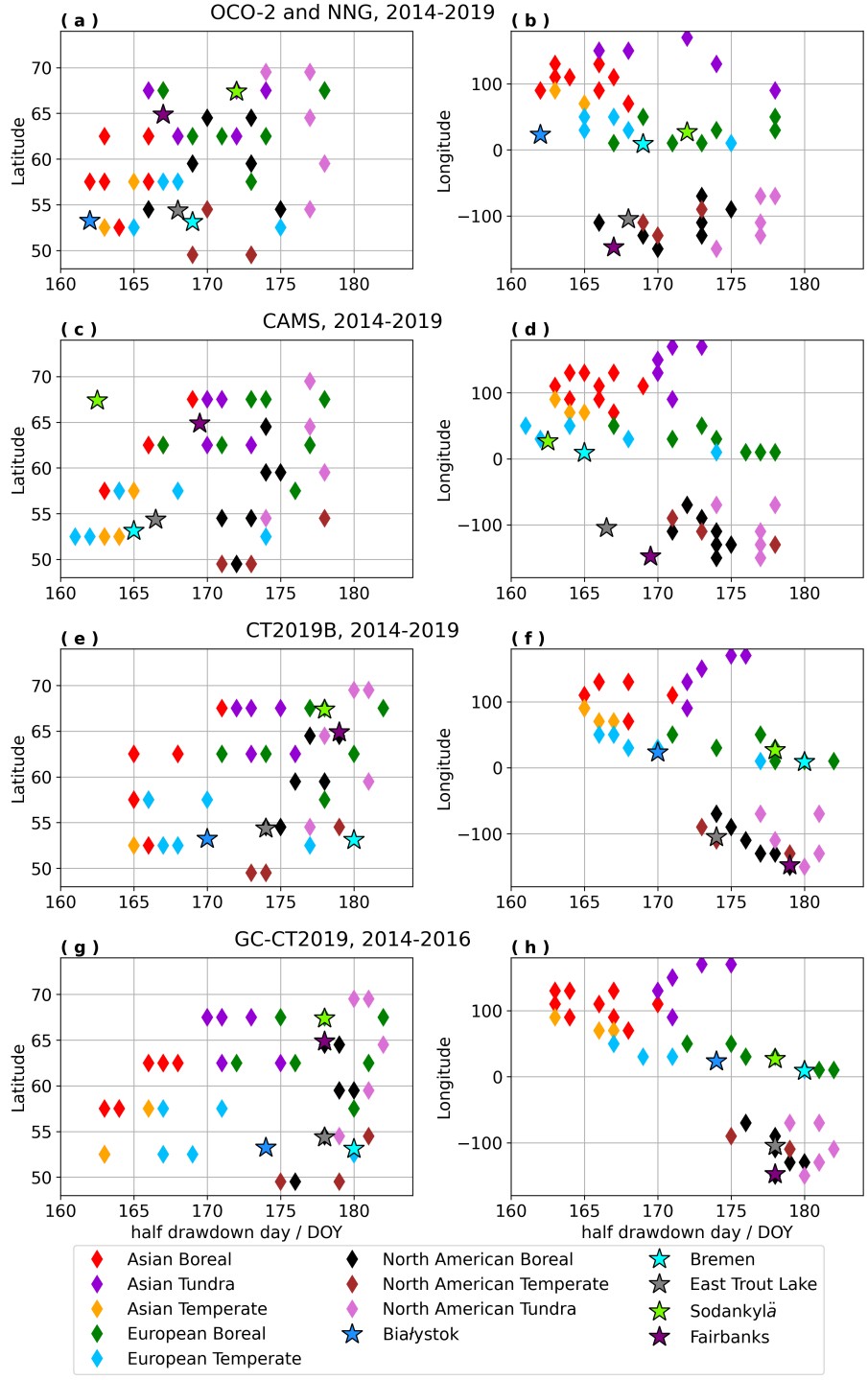

**Figure 7.** Plots of latitude and longitude correlated to HDD using observational results from OCO-2 and NNG observations ((a) and (b)), CAMS model estimates ((c) and (d)), CT2019B model estimates ((e) and (f)), and GC-CT2019 model estimates ((g) and (h)). The latitude and longitude for each zone is located at its center.

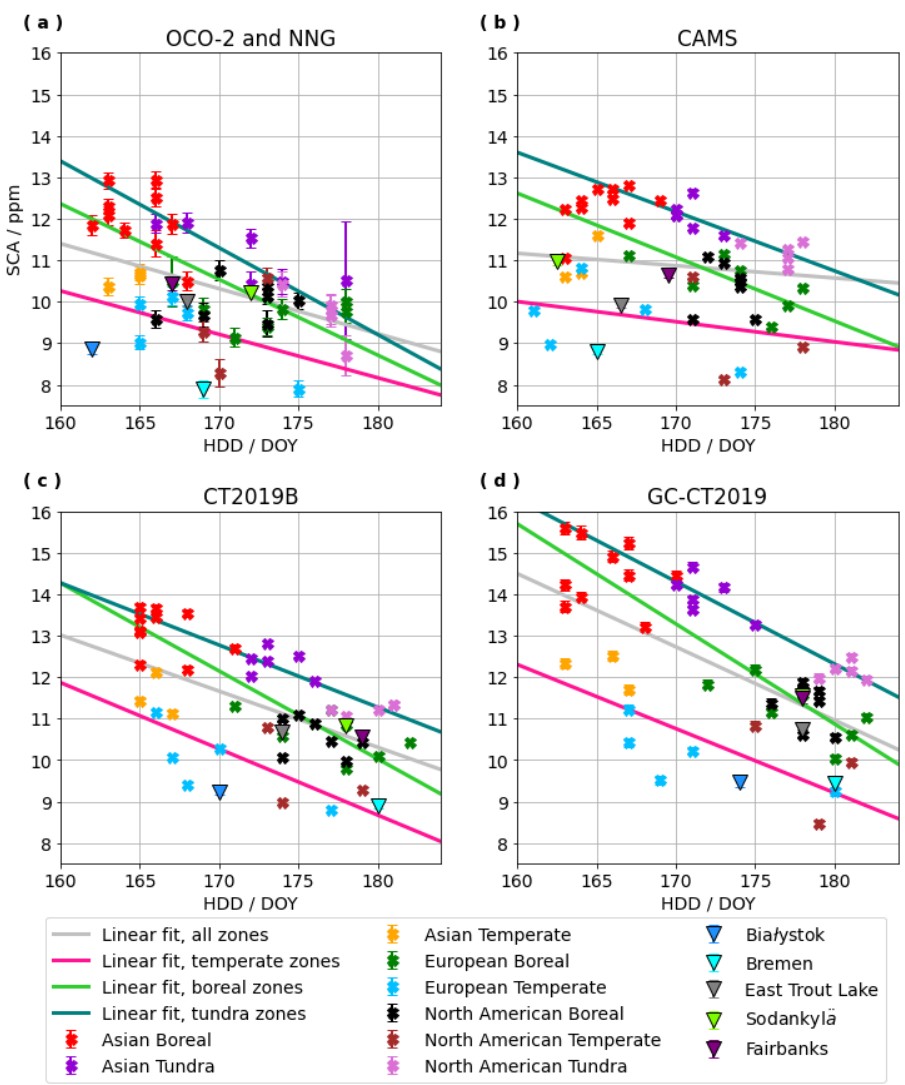

**Figure 8.** Correlations between SCA and HDD using OCO-2 and NNG seasonal cycle fits, (a), CAMS seasonal cycle fits, (b), CT2019B seasonal cycle fits (c), and GC-CT2019 seasonal cycle fits (d). Linear regressions are plotted for all zones, as well as separately for Temperate, Boreal, and Tundra regions.

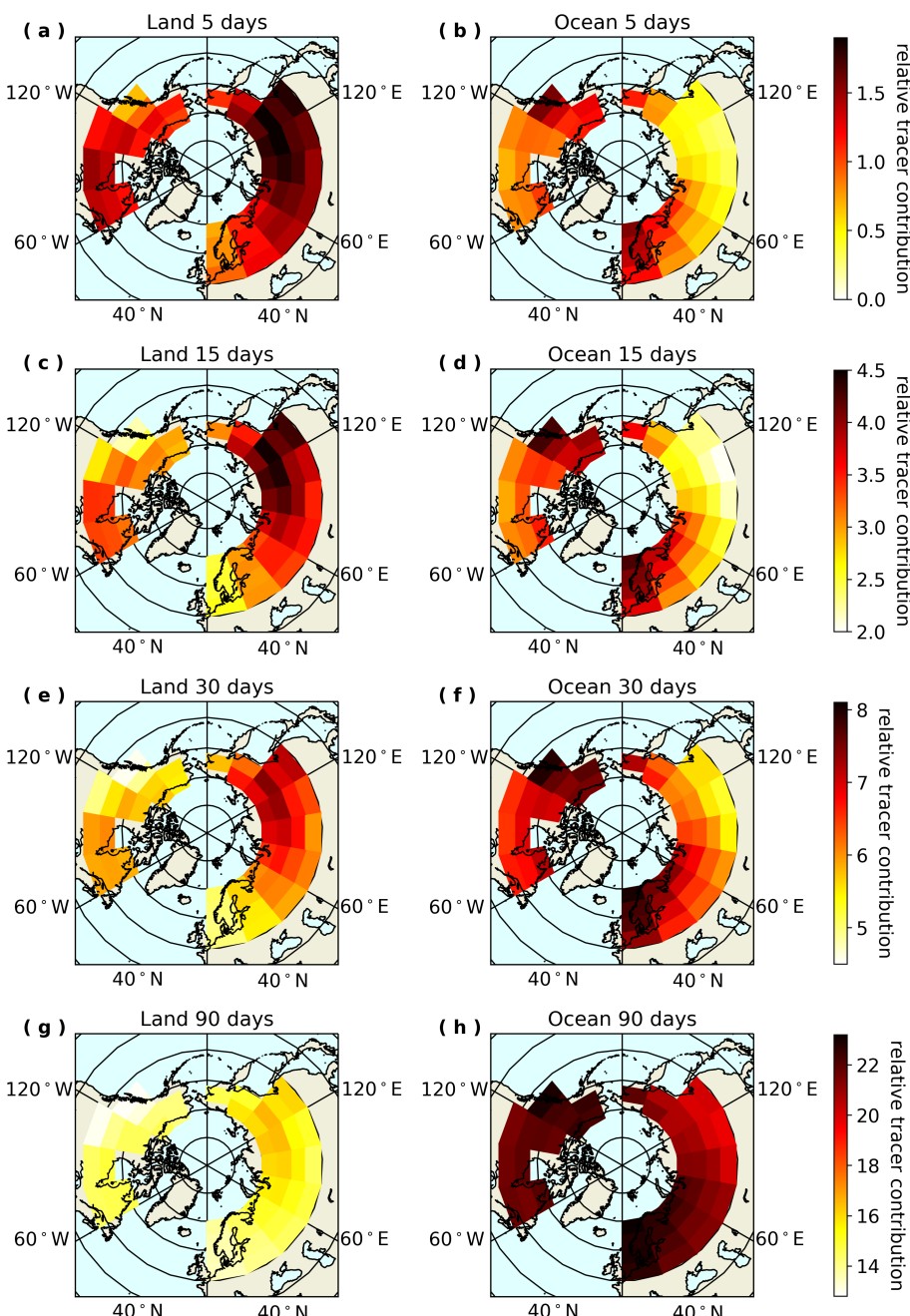

**Figure 9.** Maps of GEOS-Chem surface contact tracer contributions from land and ocean with 5, 15, 30, and 90 day lifetimes for each zone, with units that are scaled relative to an arbitrary initial release of tracer particles. Surface contact tracer contributions shown here are calculated as an average of all days in the simulation period, 2014-2016.

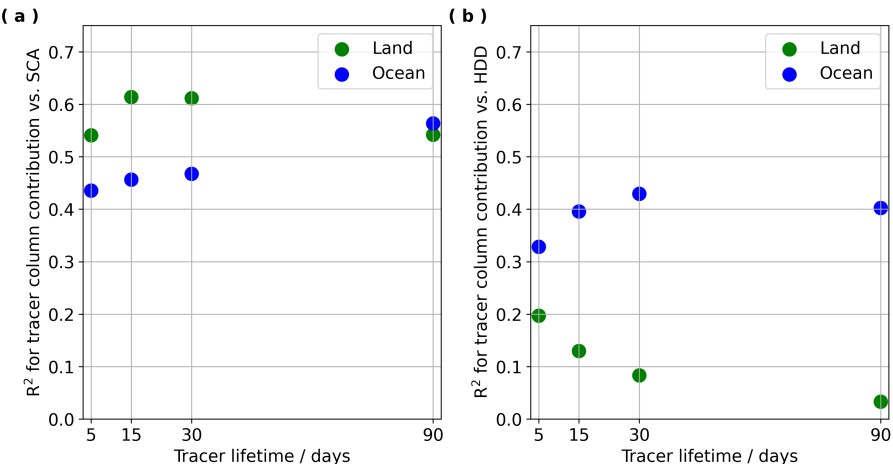

**Figure 10.** Panel (a) shows correlation coefficients for surface contact tracer contributions from land and ocean (mapped in Fig. 9) versus OCO-2 SCA (mapped in panel (a) of Fig. 4), and panel (b) shows correlation coefficients for surface contact tracer contributions from land and ocean versus OCO-2 HDD (mapped in panel (b) of Fig. 4).

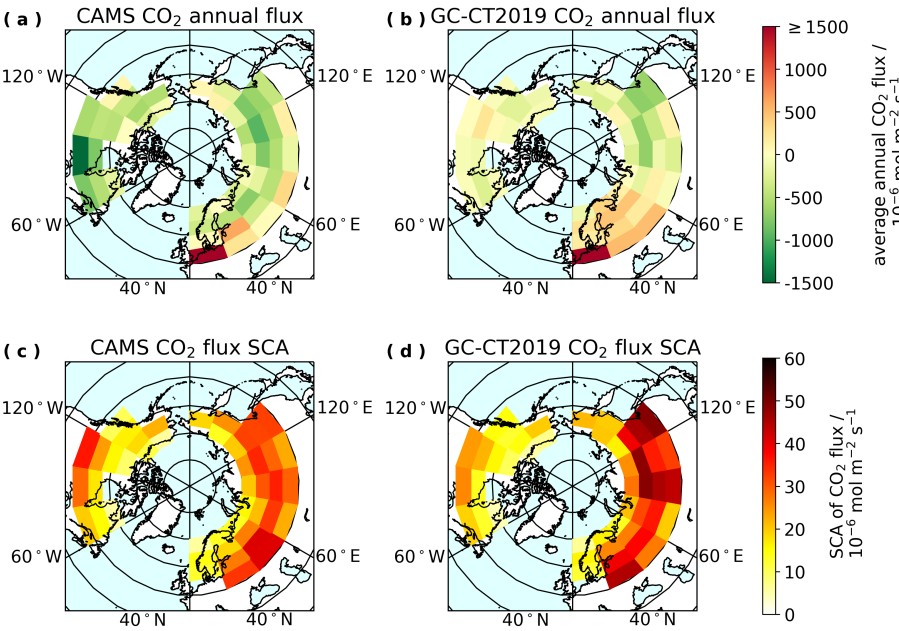

**Figure 11.** Maps of average total annual $CO_2$ flux, using GC-CT2019 (GC) and CAMS flux estimates (top), and SCA in $CO_2$ flux, calculated as the difference between the maximum and minimum of the average annual cycle in flux (bottom).

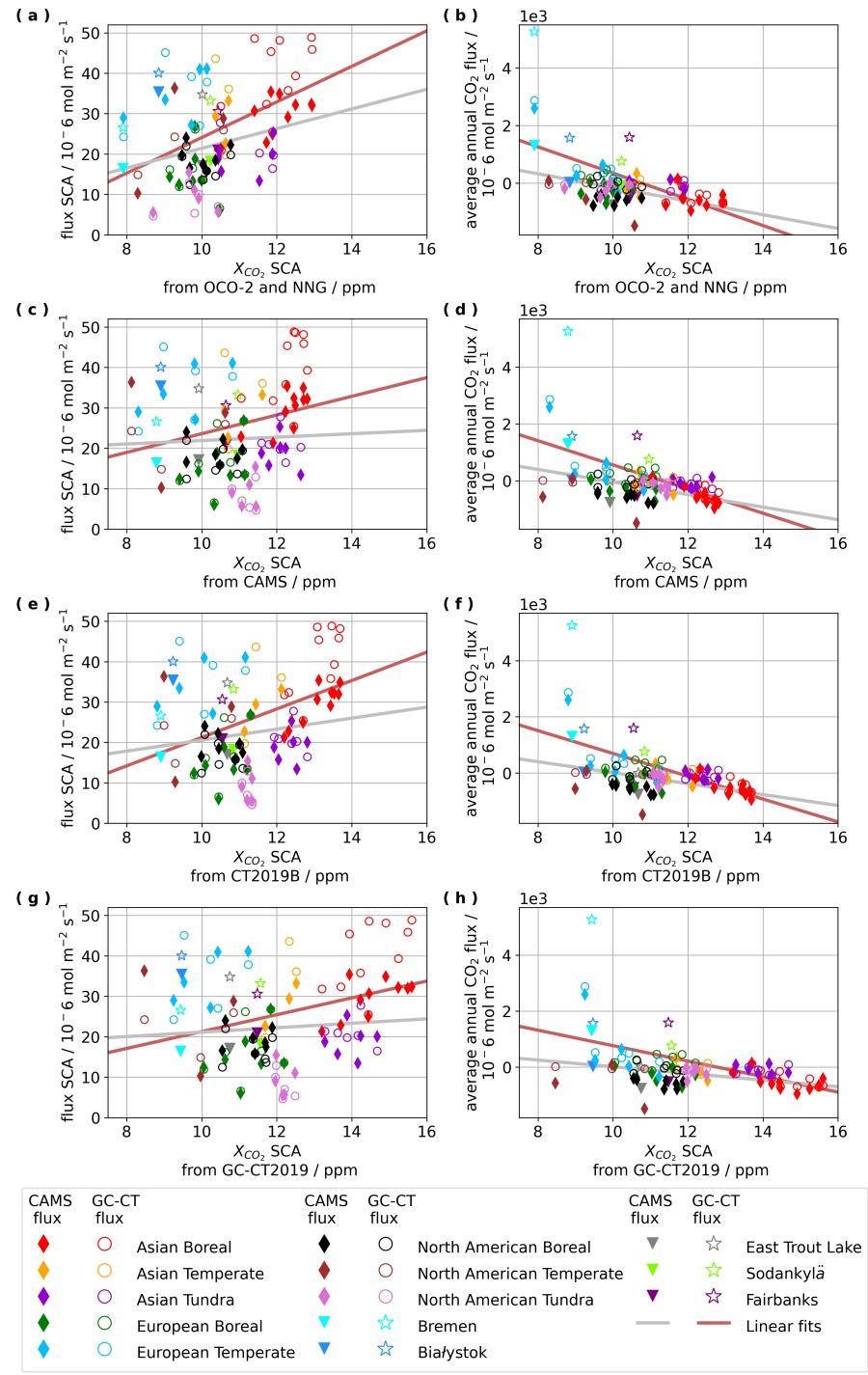

**Figure 12.** Correlation plots of flux SCA and average annual fluxes from CAMS and GC-CT2019 versus $X_{CO_2}$ SCA from OCO-2 and NNG, as well as model-derived $X_{CO_2}$ SCA from CAMS, CT2019B, and GC-CT2019.

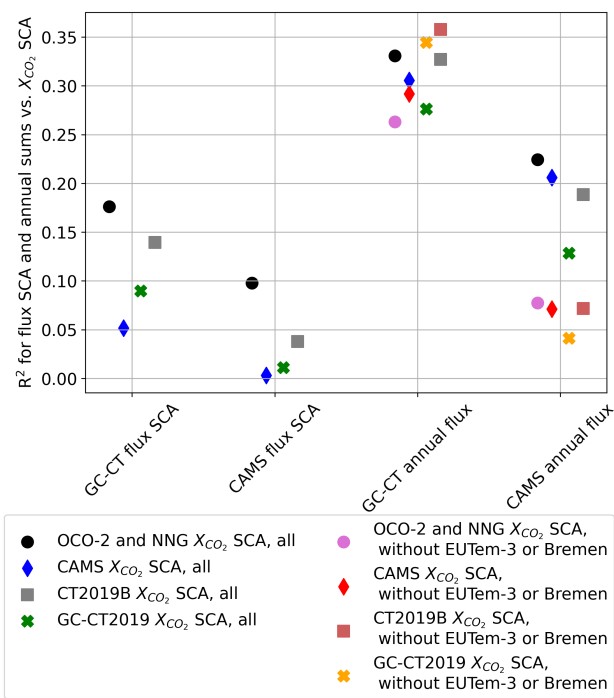

**Figure 13.** Correlation coefficients for the linear fits in Fig. 12, as well as alternative correlation coefficients for average annual fluxes versus $X_{CO_2}$ SCA with European Temperate zone 3 and Bremen removed.