# Peer review of "Spatial distributions of $X_{CO_2}$ seasonal cycle amplitude and phase over northern high latitude regions"

_Atmospheric Chemistry and Physics, 2021_

## Author Comment (AC2)

**Response to Reviewer 2**

We would like to express our gratitude to reviewer 2 for their insightful and constructive suggestions to reformat the paper and extend our analysis to additional model estimates in CarbonTracker2019. We have done our best to use these comments as a guide to composing a revised version of the paper that improves and expands on our original submission.

1 Comments and responses

5

10

15

- 1. **Comment:** My largest concern with the paper is that the authors are underutilized their contact tracing simulations. The contact tracing approach was not clearly explained, and results from these simulations were introduced only in the discussion section. This needs to be fleshed out more in the results section, with a mind to new or refined supporting figures. Fig. 10 could be modified to better guide the reader as to the similarities noted with Fig. 4. For example, could boxes be added in Fig. 10 around the zones whose amplitude is in the top Nth percentile in Fig. 4? Related to the contact tracer approach, it was unclear why the CarbonTracker posterior fluxes were run through GEOS-Chem rather than simply using the CarbonTracker posterior CO2 fields. Was this so the transport model was consistent for the CO2 simulation and the contact tracer simulation? If so, this should be explained in the text. However, the use of a single transport model is a major limitation to the authors' conclusion that accumulation of transported CO2 controls the seasonal cycle amplitude spatial patterns. It may be onerous to repeat the surface contact tracer analysis in another transport model, but could the authors combine information from the CarbonTracker (TM5) posterior and their own CarbonTracker/GEOS-Chem simulations to substantiate their conclusion?
- **Reply:** We now show results with CarbonTracker2019  $X_{CO_2}$ , as well as GEOS-Chem. The  $X_{CO_2}$  amplitudes are gener-20ally similar in GEOS-Chem and CarbonTracker/TM5, with some expected differences attributable to the different transport schemes used in GEOS-Chem and TM5. The similarity of  $X_{CO_2}$  SCA in GEOS-Chem and CarbonTracker/TM5while using the same terrestrial and ocean fluxes suggests that large-scale atmospheric transport that influences SCA is similar in the two models. We are then able to diagnose the role of transport in GEOS-Chem using the novel surface contact tracers. Furthermore, we have made the results from our contact tracer simulations a more integral part of the paper25by including dedicated methods, results, and discussion sections (see Sect. 2.7, 3.5, and 4.1 of the revised manuscript), which now provide more details on this component of the analysis.
  - 2. Comment: A second major issue is that the key insights from the research could be better laid out. Is it that the column seems to show earlier drawdown than inversions constrained by surface observations? The different behavior in Siberia compared to other zones? What are the implications for a correlation between amplitude and HDD in terms of processes that affect the seasonality of net exchange? Perhaps reorganization of the discussion section, with the surface tracer analysis presented in the results, will make it easier for these points to come through in the discussion. Reply: We have reformatted the paper, with the tracer analysis presented in the results and more streamlined discussion.

35

30

40

3. **Comment:** I struggled with Section 3.2, since the section begins with a listing of material in the supplement. The authors

know what is driving this relationship at this time.

should organize this information to provide a clear summary in the first few sentences of the paragraph/section, and then later refer to the supplementary figures for more details. In particular, the "unrealistic drop in wintertime values" might warrant a figure in the main text since the realism of the seasonal cycle fits is crucial for interpretation of the rest of the paper.

We state in that the correlation between SCA and HDD would be interesting to consider in future studies, but do not

**Reply:** This section has been revised to make it more clear. We think that it is reasonable to refer the reader to the supplement if they are interested in the exact nature of how the winter data gaps affect seasonal cycle shape, rather than including an additional figure. Some details have also been added to the methods section on seasonal cycle fitting at the

1

behest of reviewer 1 that may be pertinent to this comment.

- 4. **Comment:** The term HDD was used before the concept was introduced or defined (L29 p7). **Reply:** Text has been revised to correct this.
- 5
- 5. **Comment:** The authors cite a meridional gradient in Fig. 12, which I didn't see convincingly in any panel. This is brought up in the discussion (p14), and I don't think removing this sentence would affect the authors' underlying argument. If it is left in the document, more support is required. **Reply:** This sentence was removed.

---

## Author Response (AR1)

**Response to Reviewer 1**

We are very grateful that reviewer 1 took the time to carefully evaluate our submitted manuscript, and provide us with a valuable opportunity to reinforce the results of our analysis. We have worked to address all of the comments from reviewer 1, either in this response document or through revisions to the paper. In our efforts to address these comments, we believe that the revised manuscript is an improvement on the previous version.

**1** General Comments**

1. **Comment:** Many of the seasonal cycle fits shown in Fig. S4-S11 appear to be quite unphysical. Thus, it is unclear if the analysis is really capturing accurate SCA estimates. There should be uncertainty quantification in the SCA fits, perhaps using bootstrap resampling or another technique. Ideally, the analysis could also be performed fitting truncated Fourier series, to test the impact of the functional form on the results.

**Reply:** While we are aware that fitting seasonal cycles to a truncated Fourier series has been established in previous literature, fits of this type do not have any more physical basis than the fits that we use here, and through extensive 15 testing the fitting methods from Lindqvist et al. (2015) were found to be a better choice for this analysis. As stated in the cited literature (Lindqvist et al., 2015), these methods of seasonal cycle fitting offer improved results relative to fitting to a truncated Fourier series in situations where there are substantial gaps in the observed time-series. In particular, high latitude observations consistently have gaps in the winter-time. To get a reasonable fit to a truncated Fourier series we would need to artificially fill winter-time data, which is something we specifically want to avoid as it would detract from 20 our data-driven approach. As discussed in our alternate reply to comment 2, below, we have added a new section to the supplement that considers how removing winter-time data from CAMS time-series effects the resulting SCA (using the methods from Lindqvist et al. (2015)). We do concede that there are a few zones in the Asian Tundra region that have seemingly unphysical results, and this was discussed in section 3.2 of the manuscript. We have now also changed the map layout of the Asian Tundra zones and removed the two northern-most zones in the Asian Tundra region because these 25 zones have the largest winter gaps of any zones. The removal of these Asian Tundra zones does not seem to have changed any of the conclusions of this analysis as it pertains to the spatial distributions of  $X_{CO_2}$  seasonal cycle parameters or the roles of fluxes and atmospheric transport patterns. The Lindqvist et al. (2015) fitting methods yield fits to modelestimates that follow the data very closely, and it is our opinion that most of the seasonal fits to OCO-2 observations appear quite reasonable. This assessment of observed seasonal cycle fits is reinforced by the close agreement between spatial distributions of observed and model-derived SCA. We did evaluate some seasonal cycle fits to a truncated Fourier 30 series with a linear trend, following methods used in Wunch et al. (2013) (see Fig. R1-1), but found these fits to be far more unrealistic than the fits to a skewed sine wave presented by Lindqvist et al. (2015) (see Fig. R1-2). Even for the seasonal cycles with no data gap in the winter, there are winter-time oscillations in the fit to the truncated Fourier series that do not seem to accurately represent variability in the data, and for sites with winter data gaps the fits to the truncated Fourier series are clearly unrealistic. 35

10

**Figure R1-1.** Time-series of ground-based observations (panels (a), (b), and (c)) and satellite-based observations for the zone containing the ground site (panels (d), (e), and (f)) at two TCCON sites East Trout Lake, Canada and Sodankylä, Finland, as well as Fairbanks Alaska, where EM27/SUN observations are collected. Also shown are seasonal cycle fits for these observations to a trended truncated Fourier series of the form:  $f(t) = a_0 + a_1 t + \sum_{k=1}^{2} b_k \sin \left(2\pi k \frac{t}{365.25}\right) + c_k \cos \left(2\pi k \frac{t}{365.25}\right)$ .